



# Atmospheric oxygen as a tracer for fossil fuel carbon dioxide: a sensitivity study in the UK

Hannah Chawner [1], Karina E. Adcock [2], Tim Arnold [3,4], Yuri Artioli [5], Caroline Dylag [3], Grant L. Forster [2,6], Anita Ganesan [7], Heather Graven [8], Gennadi Lessin [5], Peter Levy [9], Ingrid T. Luijkx [10], Alistair Manning [11], Penelope A. Pickers [2], Chris Rennick [3], Christian Rödenbeck [12], and Matthew Rigby [1]

[1]School of Chemistry, University of Bristol, Bristol, UK
[2]Centre for Ocean and Atmospheric Sciences, School of Environmental Sciences, University of East Anglia, Norwich, UK
[3]National Physical Laboratory, Teddington, UK
[4]School of Geosciences, University of Edinburgh, Edinburgh, UK
[5]Plymouth Marine Laboratory, Plymouth, UK
[6]National Centre for Atmospheric Sciences, University of East Anglia, UK
[7]School of Geographical Sciences, University of Bristol, Bristol, UK
[8]Department of Physics, Imperial College London, London, UK
[9]Centre for Ecology and Hydrology, Edinburgh, UK
[10]Meteorology and Air Quality, Wageningen University and Research, Wageningen, the Netherlands
[11]Hadley Centre, Met Office, Exeter, UK
[12] Max Planck Institute for Biogeochemistry, Germany

**Correspondence:** Hannah Chawner (hannah.chawner@bristol.ac.uk), Matt Rigby (matt.rigby@bristol.ac.uk)

**Abstract.** We investigate the use of oxygen ($O_2$) and carbon dioxide ($CO_2$) measurements for the estimation of the fossil fuel component of atmospheric $CO_2$ in the UK. Atmospheric potential oxygen (APO) – a tracer that combines $O_2$ and $CO_2$, minimising the influence of terrestrial biosphere fluxes – is simulated at three sites in the UK, two of which make atmospheric APO measurements. We present a set of model experiments that estimate the sensitivity of APO simulations to key inputs:

fluxes from the ocean, fossil fuel flux magnitude and distribution, the APO baseline, and the ratio of $O_2$ to $CO_2$ fluxes from fossil fuel combustion and the terrestrial biosphere. To estimate the influence of uncertainties in ocean fluxes, we compared three ocean $O_2$ flux estimates, from the NEMO – ERSEM and ECCO-Darwin ocean models, and the Jena CarboScope APO inversion. The sensitivity of APO to fossil fuel emission magnitudes and to terrestrial biosphere and fossil fuel exchange ratios was investigated through Monte Carlo sampling within literature uncertainty ranges, and by comparing different inventory

estimates. Of the factors that could potentially compromise APO-derived fossil fuel $CO_2$ estimates, we find that the ocean $O_2$ flux estimate has the largest overall influence at the three sites in the UK. At times, this influence is comparable to the contribution to APO of simulated fossil fuel $CO_2$. We find that simulations using different ocean fluxes differ from each other substantially, with no single model estimate, or a simulation with zero ocean flux, providing a significantly closer fit to the observations. Furthermore, the uncertainty in the ocean contribution to APO could lead to uncertainty in defining an

appropriate regional background from the data. Our findings suggest that the contribution of non-terrestrial sources need to be well accounted for, in order to reduce their potential influence on inferred fossil fuel $CO_2$.





# 1 Introduction

Variations in atmospheric carbon dioxide ($CO_2$) concentrations are due to atmospheric transport and the influence of fluxes from the terrestrial biosphere, the ocean and human activities. With the ultimate aim of evaluating national emission estimates,

a major goal of several recent studies has been the isolation of only those variations due to anthropogenic fossil fuel $CO_2$ emissions. Radiocarbon, [14]C, has been widely used as a tracer for this purpose (e.g. Levin et al., 2003; Graven et al., 2009, 2018; Wenger et al., 2019), as fossil fuel emissions are fully depleted in [14]C, providing a signature with which to discriminate fossil fuel emissions from other sources and sinks. However, such measurements are expensive, they cannot be made continuously to the required precision, and in some regions there may be significant interference of [14]C emission from gas-cooled nuclear power

stations (Graven and Gruber, 2011; Bozhinova et al., 2016; Wenger et al., 2019). An alternative tracer is carbon monoxide (CO), which is released by incomplete combustion. Measurements of CO are much less expensive than those of [14]C and can be made continuously (e.g. Andrews et al., 2014; Levin and Karstens, 2007; Levin et al., 2020). However, there is large uncertainty in both the ratio of CO to $CO_2$ emissions from fossil fuel combustion, and the CO flux from non-fossil fuel sources and sinks.

Pickers (2016) and Pickers et al. (2022) show that oxygen ($O_2$) and $CO_2$ measurements, combined into Atmospheric Po-

tential Oxygen (APO) (Stephens et al., 1998), can be used as a novel tracer for fossil fuel derived $CO_2$. In their study, Pickers et al. (2022) show that their APO-derived $CO_2$ emission changes during the COVID-19 lockdowns in the UK correspond well to the changes found from bottom-up inventories. Their method, combining observations and machine-learning techniques, shows the potential of APO as a fossil fuel $CO_2$ (ff$CO_2$) tracer. The basis of this method is that the ratio of $O_2$ to $CO_2$ fluxes from the terrestrial biosphere, which are by definition removed from the $O_2$ signal through the use of the APO tracer (Stephens

et al., 1998), is relatively well constrained and invariant in space and time. For the land-based sources, $O_2$ and $CO_2$ fluxes to the atmosphere from photosynthesis, respiration, and combustion are strongly anti-correlated: $CO_2$ is taken up through photosynthesis whilst $O_2$ is released, and the reverse is true for respiration and combustion. When considering ocean fluxes, the situation is more complex, as differences in solubility (Keeling, 1988a) and carbonate chemistry (Keeling and Shertz, 1992; Keeling and Severinghaus, 2000) mean that the $O_2$ and $CO_2$ fluxes from the ocean are largely decoupled. However, previous

work has indicated that the influence of ocean fluxes on the atmospheric ratio of $O_2$ to $CO_2$ is generally smaller than the influence of fossil fuel combustion on short timescales (Pickers, 2016; Pickers et al., 2017; Chevalier and WP4 CHE partners, 2021). Pickers et al. (2017) found short-term variability in APO, $O_2$ and $CO_2$ mole fractions with only very small magnitude from the ocean when taking ship measurements.

There have been a number of promising attempts to incorporate $O_2$ modelling as a tracer for ff$CO_2$. Kuijpers et al. (2018)

modelled $O_2$ for the autumn of 2015 for three sites in the UK and the Netherlands, finding good agreement with observations. APO modelling was investigated to derive European ff$CO_2$ fluxes by several groups within the $CO_2$ Human Emissions project (CHE, work package 4, Marshall et al., 2019; Chevalier and WP4 CHE partners, 2021). Comparing with results from $\Delta^{14}CO_2$ and CO modelling, they found that APO-derived ff$CO_2$ gave the strongest correlation to direct ff$CO_2$ models using STILT and TNO fluxes. The APO models were affected by oceanic fluxes at some coastal sites, although for most coastal sites the ocean

influence, using ocean fluxes from NEMO - PlankTOM5, was considerably smaller than that of the ff$CO_2$.



Two measurement sites equipped with high-frequency $CO_2$ and $O_2$ instruments have been established in the UK, one at Weybourne Atmospheric Observatory (WAO) in the East of England and one at Heathfield tower (HFD) in the South of England. In this paper, we perform simulations of $CO_2$ and $O_2$ focusing on these locations, along with a third site at Ridge Hill (RGL). Although $O_2$ measurements are not available from RGL, it is included to examine the modelled APO further inland. We test the sensitivity of the APO simulation to changes in a set of uncertain model input parameters, to determine whether a robust tracer of national scale fossil fuel $CO_2$ can be derived.

## 1.1 Modelling Atmospheric Potential Oxygen

As $O_2$ is abundant in the atmosphere, dilution by trace gases can have a non-negligible effect on its mole fraction which may erroneously be attributed to an $O_2$ flux. To minimise this influence, oxygen measurements are commonly reported as a ratio with respect to the atmospheric nitrogen mole fraction as $\delta(O_2/N_2)$ (Keeling and Shertz, 1992):

$$\delta(O_2/N_2) = \frac{(O_2/N_2)_{sample} - (O_2/N_2)_{reference}}{(O_2/N_2)_{reference}} \times 10^6 \tag{1}$$

where $(O_2/N_2)_{sample}$ is the $O_2/N_2$ ratio of a sample, and $(O_2/N_2)_{reference}$ is from a reference gas cylinder. $\delta(O_2/N_2)$ is expressed in per meg.

We can define the tracer APO(e.g. Stephens et al., 1998; Gruber et al., 2001; Battle et al., 2006) that is largely unaffected by exchanges with the terrestrial biosphere, but sensitive to fossil fuel and ocean fluxes. This is a weighted combination of $O_2$ and $CO_2$ which isolates the oceanic and fossil fuel components:

$$APO = O_2 + \alpha_B \times (CO_2 - 350) \tag{2}$$

where $APO$ is a mole fraction, $\alpha_B$ is the $O_2$:$CO_2$ exchange ratio for the land biosphere, $O_2$ and $CO_2$ are the atmospheric mole fractions of $O_2$ and $CO_2$ respectively, and 350 ($\mu$mol mol$^{-1}$) is an arbitrary reference.

Equations 1 and 2 can be combined, expressing APO in units of per meg (Stephens et al., 1998):

$$APO = \delta(O_2/N_2) + \left(\frac{\alpha_B}{S_{O_2}}\right) \times (CO_2 - 350) \tag{3}$$

where $S_{O_2}$ is the standard mole fraction of $O_2$ in air, equal to 0.20946 (Machta and Hughes, 1970).

### 1.1.1 The regional contribution to atmospheric APO

The regional contribution of atmospheric APO can be estimated by combining the mole fraction contribution of $O_2$, $CO_2$, and $N_2$. Following the derivation in Manning and Keeling (2006), the deviation of APO can be expressed as:

$$\Delta(\delta APO) = \frac{Z + (\alpha_F - \alpha_B)F + \alpha_B O}{S_{O_2}(1 - S_{O_2})} - \frac{N}{S_{N_2}} \tag{4}$$

$$= \frac{Z + F_O - \alpha_B F + \alpha_B O}{S_{O_2}(1 - S_{O_2})} - \frac{N}{S_{N_2}} \tag{5}$$



where $Z$ and $O$ respectively are the $O_2$ and $CO_2$ mole fraction contributions from the ocean; $F$ and $F_O$ are the contributions of $CO_2$ and $O_2$ respectively from fossil fuel combustion and cement production; $N$ is the $N_2$ contribution; $\alpha_F$ and $\alpha_B$ are the fossil fuel and biospheric exchange ratios; and $S_{N_2}$ is the mole fraction of $N_2$ in dry air, given as 0.78084 (Weast and Astle, 1982), where this and $S_{O_2}$ are used to convert from units of ppm ($\mu$mol/mol) to per meg. A correction of $(1 - S_{O_2})$ accounts for dilution effects of $O_2$ (Kozlova et al., 2008).

When estimating the exchange of $N_2$ we need only to consider the ocean contribution as the other components are assumed to be negligible (Ciais et al., 2007). We assume a constant value for $\alpha_B$ for the UK of $-1.07 \pm 0.04$ (Marshall et al., 2019; Pickers). $\alpha_F$ varies for different fuel types, having values of -1.17 for coal, -1.44 for oil, -1.95 for gas, and 0 for cement production (Keeling, 1988b; Steinbach et al., 2011), and can be estimated for the UK by combining fossil fuel emissions estimates and fuel usage statistics, as outlined in Section 2.2.2. Variations in $\alpha_F$ are not well studied or constrained, however we follow Jones et al. (2021) in assuming an uncertainty of $\pm 3$ per cent.

## 2 Methodology

### 2.1 Observations

At both the measurement sites, WAO and HFD, atmospheric $O_2$ measurements are made using 'Oxzilla' lead fuel cell analysers (Sable Systems International Inc.) placed in series with non-dispersive infrared (NDIR) $CO_2$ 'Ultramat 6E' analysers (Siemens Corp.). The gas handling for each system is similar to that of Adcock et al. (in prep.), Pickers et al. (2017) and Stephens et al. (2007), to ensure stable pressures and flow rates are maintained and to avoid $O_2/N_2$ fractionation effects. A two-stage drying system (Wilson, 2013; Barningham, 2018; Adcock et al., in prep.) reduces the dew point of the sample air to approximately -90°C. Calibration gases, consisting of secondary standards that are stored horizontally in thermally insulated enclosures, are used to characterise analyser responses on the WMO $CO_2$ scale maintained by NOAA and the Scripps Institution of Oceanography scale for $O_2$, by employing routines and protocols similar to those of Kozlova and Manning (2009).

Weybourne Atmospheric Observatory (WAO; https://weybourne.uea.ac.uk/) is a coastal measurement station in Norfolk, in the east of England (52°57'02"N, 1°07'19"E) which has been routinely sampling $CO_2$ and $O_2$ since May 2010. Established in 1992, WAO is a Global Atmospheric Watch (GAW) Regional station, an National Centre for Atmospheric Sciences (NCAS) Atmospheric Measurement Facility (AMF), and an Integrated Carbon Observation System (ICOS) Class 2 station. Air is alternately sampled from two identical aspirated inlets at 15 magl (Blaine et al., 2006).

Heathfield (HFD) is a tall-tower measurement site that is part of the UK Deriving Emissions linked to Climate Change (DECC) network (Stanley et al., 2018) which has been sampling $CO_2$ and $O_2$ since June 2021. The site is in an agricultural area in the south of England (50°58'36.3"N, 0°13'49.728"E), around 25 km north of the English Channel. Air is alternately sampled from two identical aspirated inlets (Blaine et al., 2006) at 100 magl.

Ridge Hill is also a tall-tower measurement site in the UK DECC network in Herefordshire (51°59'50.766"N, 2°32'23.64"W). Although $CO_2$ is sampled here, $O_2$ is not, yet we include this site in the analysis to test the model at a more inland UK site.



The repeatability of the $O_2$ measurements from Weybourne, which is determined from regular measurements of a target tank, typically ranges from $1.68 \pm 1.09$ per meg to $3.31 \pm 5.46$ per meg (Adcock et al, in prep). This exceeds WMO repeatability goals (WMO, 2019) for $O_2$, but is nevertheless amongst the most precise globally. The repeatability is calculated using the method explained in Pickers et al. (2017) and is reported with $\pm 1\sigma$ uncertainty to represent how the measurement system repeatability varies over time. During the period February to November 2015, the $O_2$ measurement repeatability was significantly larger $(10.71 \pm 10.45)$ than usual, caused by poor performance of the Oxzilla analyser. As described in Section 2.2, we model 2015 as it is the most recent year for which outputs exist for all of the ocean models used. This larger repeatability does not significantly affect the accuracy of the $O_2$ measurements, but does compromise the detection limit, meaning that smaller synoptic variations in APO ($<10-20$ per meg) may be masked during this period by the measurement imprecision. $CO_2$ repeatability was not affected, and is $0.005 \pm 0.023$ ppm on average at Weybourne, calculated from over 8000 target tank measurements made from $2010-2021$.

## 2.2 Modelling APO

We use a Lagrangian particle dispersion model (LPDM) to simulate APO at the measurement sites in the south of the UK. The key components of our simulation are the LPDM "footprints", a set of flux estimates, and boundary conditions at the edge of our domain. The following sections outline how each component was produced and used.

For our analysis we focus on the year 2015, chosen because time-resolved ocean model outputs are available for all ocean models considered here, described in Section 2.2.2. Weybourne measurements are available for 2015 and are compared to the simulation in Section 3. Heathfield observations are only available from June 2021, when time-resolved ocean fluxes are not available, so model outputs, derived using climatological fluxes, are compared to the observational data for this site and shown in the Supplement. Simulations at Ridge Hill are shown in the Supplement.

We also model the total $CO_2$ and $O_2$ mole fraction at Weybourne to compare the correlations with those observations to the equivalent for APO.

### 2.2.1 The Atmospheric Model

Simulations of atmospheric transport and dispersion are carried out using the Numerical Atmospheric-dispersion Modelling Environment (NAME III, version 7.2), the UK Met Office's LPDM (Jones et al., 2007). NAME was run in time-reversed mode, in which we tracked thousands of model particles back in time for 30 days from observation sites (see e.g. Manning et al., 2011). The motion of hypothetical "particles" is simulated based on meteorological fields from the Met Office Unified Model analyses (Cullen, 1993). The "footprint" of each measurement was estimated by recording locations and times at which particles interacted with the Earth's surface (defined as being the lowest 40 m of the atmosphere in this case). These footprints define the sensitivity of mole fractions at a measurement site to the flux from each grid cell in the domain. Our domain covered most of Europe, the east coast of North and Central America, and North Africa, extending across the longitude/latitude range: $10.729-79.057°$N and $97.9°$W - $39.38°$E (shown in Supplementary Figure S1). The footprints have the resolution $0.234°$ by $0.352°$ (roughly 25 km by 25 km over the UK).



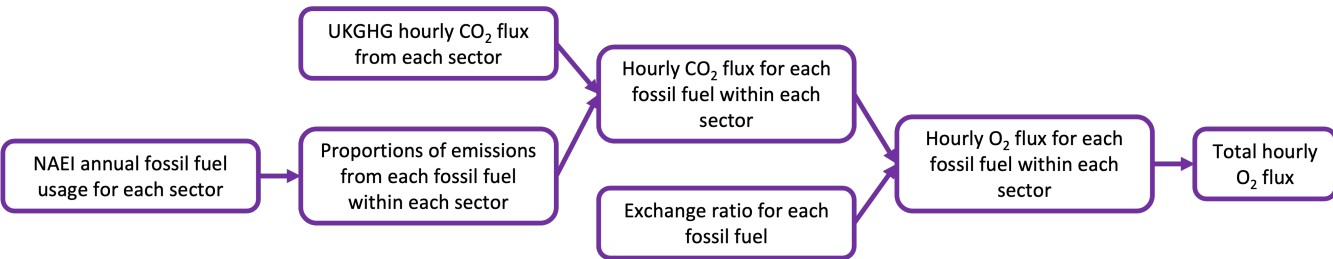

**Figure 1.** Calculation of UK fossil fuel $O_2$ fluxes from $CO_2$ flux estimates and fuel usage statistics from the UK National Atmospheric Emissions Inventory (NAEI), where flux estimates are downscaled to an hourly resolution using UKGHG (White et al., 2019).

The NAME footprints used for this study are disaggregated in time with the method described by White et al. (2019). To account for the influence on the mole fractions of rapid variations in $CO_2$ flux, footprints are generated hourly for the 24 hours preceding a simulated data point. Time-integrated footprints are then used for the remaining 29 days of the simulation. The modelled regional contribution to the mole fraction of a species, $Y_t$, at a time-step, $t$, can then be estimated by combining the flux field with the high-time-resolution NAME footprint, as shown by equation 6 (White et al., 2019):

$$Y_t = \sum_{h=0}^{H} \sum_{j=0}^{n} fp_{t-h,j} \times q_{t-h,j} + \sum_{j=0}^{n} fp_{remainder_j} \times q_{month_j} \tag{6}$$

where $H$ is the number of hours back in time over which the footprint is disaggregated, for which we use 24; $h$ is the number of hours back in time before the particle release time, $t$; $j$ is the grid cell and $n$ is the maximum number of grid cells; $fp_{t-h,j}$ is one grid cell of the footprint for that time; $q_{t-h,j}$ is one grid cell of the flux field; $fp_{remainder_j}$ is the remaining 29-day footprint; and $q_{month_j}$ is the monthly average flux for the grid cell (by calendar month). White et al. (2019) discusses this method in more detail, including the effects of varying the level of time-disaggregation of the footprint, $H$.

### 2.2.2 Flux products

We model the regional contribution to APO separately for each of the components of Equation 5 ($Z$, $F_O$, $F$, $O$), using Equation 6 to combine the flux estimates and NAME footprints. Here we describe how the fluxes for each component are estimated.

Anthropogenic $CO_2$ flux estimates for the UK are taken from the UK National Atmospheric Emissions Inventory (NAEI), where estimates at a downscaled hourly resolution are derived using the UKGHG model (White et al., 2019). Outside of the UK, anthropogenic flux estimates from EDGAR (Emissions Database for Global Atmospheric Research) are used. As NAEI includes the anthropogenic $CO_2$ flux estimates from both fossil fuel and non-fossil fuel sources (e.g. peat and biomass), we use the method described in Figure 1 and Equations 7 and 8 to remove emissions associated with non-fossil fuel sources, and thus estimate the fossil fuel UK $CO_2$ and $O_2$ flux:

$$CO_{2ff} = \sum_s \sum_e CO_{2s} R_{se} \tag{7}$$



$$O_{2ff} = \sum_s \sum_e CO_{2s} R_{se} \alpha_{fe} \tag{8}$$

where $s$ is the SNAP sector (Selected Nomenclature for reporting of Air Pollutants, see e.g. Tsagatakis et al., 2022), $e$ is the fuel or source type (coal, oil, gas, non-combustion, or cement production), $CO_{2s}$ is the $CO_2$ flux for the sector, $R_{se}$ is the proportion of $CO_2$ emissions within the SNAP sector associated with the fuel type, and $\alpha_{fe}$ is the fossil fuel exchange ratio for the fuel type. We use NAEI statistics of the annual fuel usage for each SNAP sector[1] to determine $R_{se}$, assuming that the 

170 ratio of fuels used within each sector is constant throughout the year. When determining the fuel type associated with NAEI emissions estimates we follow the assumptions given by Jones et al. (2021), that emissions from the non-energy use of fuels and solvent sector relate to non-combustion use of oil, and emissions from the production of non-metallic minerals relate to cement clinker production. Using the exchange ratio for each fuel, $\alpha_{fe}$, we then convert from $CO_2$ to $O_2$ flux for each fuel within each sector, and take the sum to give the total hourly $O_2$ flux throughout the year. The $O_2$ flux from outside of the UK 

is estimated using EDGAR $CO_2$ fields and $\alpha_F$ estimates from GridFED (Jones et al., 2021).

We compare ocean $CO_2$ and $O_2$ fluxes derived from NEMO – ERSEM simulations (NE, Butenschön et al., 2016; Madec and NEMO System Team, 2022), the ECCO – Darwin model (ED, Carroll et al., 2020) and the Jena CarboScope APO inversion (JC, Rödenbeck et al., 2008), as well as a model with ocean fluxes excluded. All of the ocean fluxes have daily time resolution and raw spatial resolutions of $0.199° \times 0.333°$, $2.0° \times 2.5°$, and $0.066° \times 0.110°$ for ED, JC, and NE respectively, which are 

180 regridded to match the NAME spatial resolution for our analysis.

ED determines ocean-atmosphere transfer of $O_2$ and $CO_2$ by combining the $CO_2$ partial pressure difference across the air-sea interface with the relationship between wind speed and gas transfer, as described by Wanninkhof (1992). The Darwin Project biogeochemical model resolves the cycling of $CO_2$ and $O_2$ and its ocean ecology includes phytoplankton and zooplankton (Brix et al., 2015; Carroll et al., 2020). JC estimates $CO_2$ and APO fluxes using a Bayesian atmospheric inversion and mea- 

185 surements from 23 $CO_2$ stations and up to 10 $O_2$ stations (including Weybourne, Rödenbeck et al., 2003, 2008, 2018). For the JC APO inversion oceanic $CO_2$ fluxes are estimated from the interpolation of pCO$_2$ data, Air-sea fluxes of $O_2$ and $CO_2$ in NE are calculated starting from the gradient of those gases between the atmosphere and the water and using Nightingale et al. (2000) to estimate the gas transfer coefficient. The concentration of $O_2$ and $CO_2$ in the water are the results of dynamical processes in the ecosystem represented in the model, and in particular photosynthesis from phytoplankton and respiration of all 

190 planktonic community as well as benthic organisms. More details on the dynamics of these gases can be found in Butenschön et al. (2016). For all of our APO models we use a nitrogen flux field estimated from NEMO heat fluxes by Equation 9:

$$q_{ocean_N} = -\frac{dC_{eq}}{dT} \frac{\dot{Q}}{C_p} \tag{9}$$

---

[1]https://naei.beis.gov.uk/data/data-selector





where $dC_{eq}/dT$ is the temperature derivative of the solubility, $\dot{Q}$ is the ocean heat flux (positive for transfer from the ocean to the atmosphere), and $C_p$ is the heat capacity of seawater (Keeling et al., 1993). $dC_{eq}/dT$ is estimated using:

$$\ln C = A_0 + A_1 T_S + A_2 T_S^2 + A_3 T_S^3 + S(B_0 + B_1 T_S + B_2 T_S^2) \tag{10}$$

with

$$T_S = \ln\left(\frac{571.3 - T}{T}\right) \tag{11}$$

where $C$ is the gas concentration, $T$ is the temperature (K), $S$ is the salinity and the $A$ and $B$ coefficients are defined in Hamme (2004). The surface heat flux is calculated by NEMO as the balance between the non-solar heat (sum of sensible, latent and long wave heat fluxes) and the incoming solar radiation (Madec and NEMO System Team, 2022). Both the ocean temperature and salinity are derived from the NE simulation.

When modelling $CO_2$ and $O_2$ mole fractions separately, we must include a terrestrial flux component. For this we use $CO_2$ flux estimates from the Organising Carbon and Hydrology In Dynamic Ecosystems (ORCHIDEE, Krinner et al., 2005) model. ORCHIDEE is a dynamic vegetation model which simulates the principal biospheric processes influencing the global carbon cycle, including photosynthesis, autotrophic and heterotrophic respiration. To estimate the terrestrial $O_2$ flux we multiply the $CO_2$ flux by $\alpha_B$, which we assumed is equal to $1.07 \pm 0.04$ (see Section 1.1).

### 2.2.3 APO boundary conditions

With the method of Lunt et al. (2016), we model the contribution from the boundary conditions at the edge of our domain using global atmospheric fields of APO mixing ratios from the JC global APO inversion (Rödenbeck et al., 2008, version apo99X_WAO_v2021). These boundary conditions are propagated to the measurement site by tracking the location at which NAME model particles leave the domain, thus providing a baseline estimate at the site. The baseline estimated from the boundary conditions is adjusted for consistency with the observations. To do this, we adjust the JC background for each month such that the simulated APO during periods of minimal terrestrial influence (defined as the 90 percentile of APO in a simulation with no ocean fluxes) are consistent with the observations at the same times. The original and adjusted JC backgrounds are shown in Figure S2 in the Supplement.

### 2.3 Sensitivity experiments

Model simulations of APO are sensitive to uncertainties in several inputs of Equation 5. In this section, we outline how we investigate the sensitivities to the biospheric and anthropogenic exchange ratios ($\alpha_B$ and $\alpha_F$), ocean fluxes, fossil fuel $CO_2$ emissions, baseline, and atmospheric model.





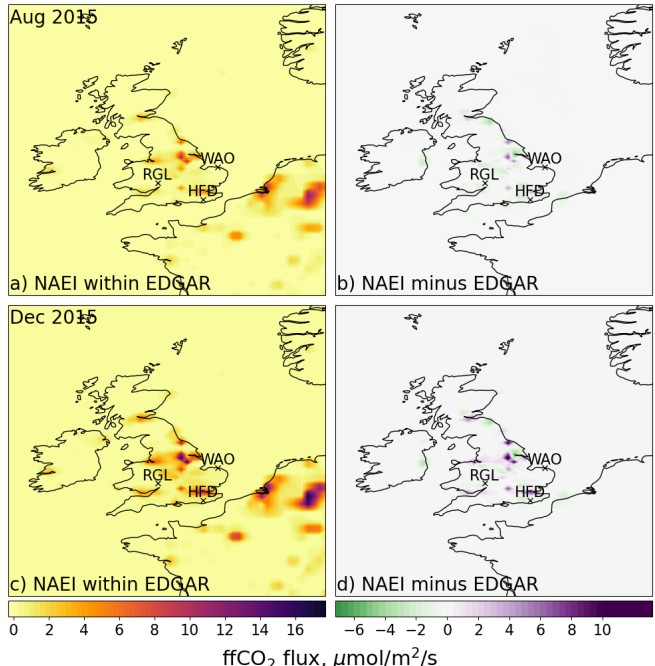

**Figure 2.** The ffCO$_2$ flux estimated by NAEI, embedded in EDGAR *(panels a and c)*, and the difference between the NAEI and the EDGAR fields *(panels b and d)* for August *(panels a and b)* and December 2015 *(panels c and d)*. By definition panels *b* and *d* are zero outside of the UK. The crosses show the locations of the sites included in this study: HFD, RGL, and WAO.

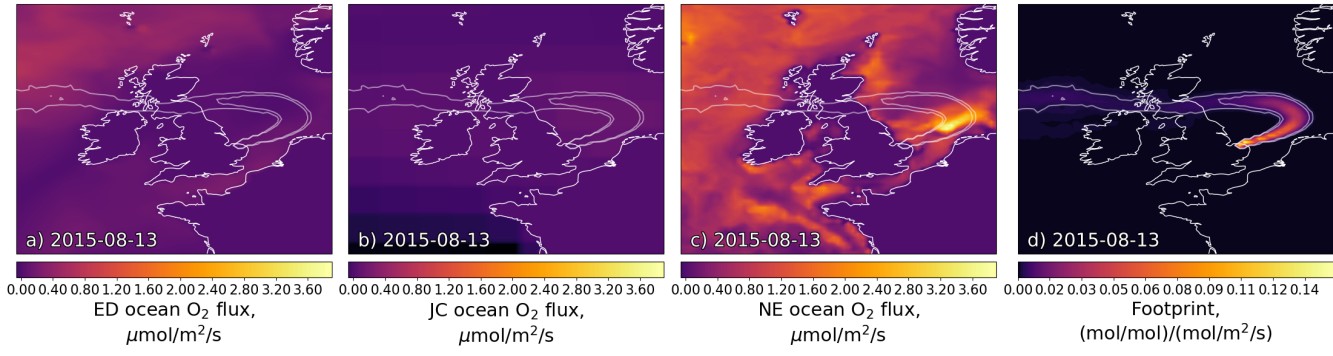

**Figure 3.** The daily mean O$_2$ ocean flux fields from the ED model *(panel a)*, the JC Inversion *(panel b)* and NE model *(panel c)*, and the NAME footprint *(panel d)* on the 13/08/2015, at a time at which the ED and NE ocean fluxes dominate the simulated APO and when there is a large difference between the estimated O$_2$ contribution from the three flux estimates. The flux fields have the 0.002 and 0.005 (mol/mol)/(mol/m$^2$/s) footprint contour overlaid.



### 2.3.1 Sensitivity to the exchange ratios: $\alpha_B$ and $\alpha_F$

To investigate our sensitivity to $\alpha_B$ and $\alpha_F$ in Equation 5 we employ a Monte Carlo method, randomly generating a value for each from a Gaussian distribution with a standard deviation of 0.04 mol/mol (Marshall et al., 2019) and 3 per cent (Jones et al., 2021) for $\alpha_B$ and $\alpha_F$ respectively. Doing so, we generate 1000 values for the APO time-series.

As $\alpha_F$ varies for different fuels we must take this into account when studying the sensitivity to $\alpha_F$. As described in section 2.2.2, the fossil fuel $O_2$ flux for each sector is calculated using $\alpha_F$ based on the proportion of fuels consumed within that sector. We therefore initially investigate the sector-wise sensitivity of the $O_2$ flux to $\alpha_F$ for each fossil fuel: coal, oil, and gas. Then we combine this information to determine the overall sensitivity of the fossil fuel $O_2$ flux and the APO simulation to $\alpha_F$.

### 2.3.2 Sensitivity to fossil fuel flux magnitude and distribution

The modelled APO is dependent on fossil fuel flux estimates, and here we study the sensitivity of the modelled fossil fuel contribution to the atmospheric concentration of $CO_2$ and $O_2$. We first examine how the APO model may be affected by estimates of the distribution of ff$CO_2$ emissions. As shown in Figure 2, there are differences in the distribution of $CO_2$ flux estimated by two different $CO_2$ inventories: NAEI and EDGAR, and we can compare the APO model using these to investigate this effect. As discussed in Section 2.2, our APO model uses NAEI ff$CO_2$ emissions estimates for the UK which are embedded in those of EDGAR, as well as NAEI fuel usage statistics to calculate ff$O_2$ uptake. Here we compare with EDGAR $CO_2$ emissions, using the GridFED estimates of $\alpha_F$ to estimate ff$O_2$.

We further investigate the sensitivity of the model to the magnitude of ff$CO_2$ estimates using a Monte Carlo ensemble in which the overall $CO_2$ flux in the entire domain is allowed to vary by $\pm10\%$ (considerably larger than the difference between EDGAR and the NAEI, which is approximately 0.7%, but chosen so that the effect on APO can be readily identified).

### 2.3.3 Sensitivity to ocean flux

Figure 3 shows the ocean flux fields from the ED and NE models and the JC inversion. For illustration, this figure is shown for a period (13[th] August 2015) when the footprint for WAO is predominantly across the ocean. On this date, and in general, there is a much larger flux in coastal regions in the NE ocean model compared with both the ED and JC estimates. Unlike exchange ratios, the sensitivity of simulated APO to ocean fluxes cannot readily be described by an uncertainty on a single parameter. Therefore, to examine the sensitivity to this term we produce APO timeseries using the three different flux estimates such that we can qualitatively compare the effect on APO magnitude and variability, and compare the correlation of each model with the observations. We also produce a timeseries with the ocean component excluded to examine whether the fit to the observations can be improved by assuming a negligible ocean contribution.

### 2.3.4 Sensitivity to the background estimate

We study the effects of the background APO estimate on our simulations. The background represents the APO variability that is representative of the well–mixed atmosphere at the UK's latitude, excluding local influences. To do so, here we compare the



modelled $\Delta(\delta APO)$ (calculated using equation 5) with background-subtracted observations at Weybourne throughout 2015. We compare two methods to subtract the background from the observations. First we estimate a baseline from the APO observations using the 'REBS' statistical fitting routine (Robust Extraction of Baseline Signal, Ruckstuhl et al., 2012; Pickers et al., 2022) with a span value of 0.03, equivalent to a smoothing window of approximately one week. This smoothing window

was thought to be the most appropriate for incorporating wider-scale APO signals from outside Europe into the background term while simultaneously excluding local influences. For our second background subtraction we use the JC background estimate, estimated from boundary conditions propagated to the measurement site using NAME (Section 2.2.3). A monthly adjustment is made to the JC background to account for offsets observed in some months, as described in Section 2.2.3. This gives us two estimates of observation-derived ffCO$_2$, using which we can compare the background subtraction method.

These background estimates are inherently different: for example the REBS baseline incorporates regional ocean seasonality whereas the JC estimate represents contributions from outside of the domain. However, comparing both background subtractions gives us an idea of the impact of differences between background estimates, such as their variability.

### 2.3.5 Sensitivity to the Atmospheric Model

As discussed in Section 2.2, in this study we use the NAME atmospheric transport model. Although NAME has been exten-
sively inter-compared to other transport models in several publications (e.g. Brunner et al., 2017; Rigby et al., 2019; Monteil et al., 2020), systematic errors in NAME will influence the comparison with observations. Whilst an extensive model inter-comparison exercise is beyond the scope of this paper, to provide a simple comparison with another widely used modelling system, we compare the NAME fossil fuel CO$_2$ time series to that of CarbonTracker Europe (CTE2022, van der Laan-Luijkx et al., 2017; Friedlingstein et al., 2022). CTE2022 uses the TM5 transport model (Krol et al., 2005) driven by ERA-5 meteorol-
ogy to transport prior fluxes globally, and surface CO$_2$ fluxes are optimized on a weekly timestep over the period $2000 - 2021$. The prior fluxes are from the SiB4 biosphere model (Haynes et al., 2019), GFAS fire emissions (Kaiser et al., 2012), GridFED fossil fuel emissions (Jones et al., 2021) and JC ocean fluxes. CO$_2$ mole fractions based on the optimized CTE2022 at WAO are used here, with separate tracers are available for each of the described flux components.

### 2.4 Fossil fuel CO$_2$ mole fraction

Previous studies have indicated that we can assume that ocean fluxes do not contribute strongly to the overall APO at a measurement site over short time scales (Pickers, 2016; Pickers et al., 2017; Chevalier and WP4 CHE partners, 2021). Based on this assumption, it has been proposed that we can estimate regional ffCO$_2$ mole fractions from APO, following Pickers (2016):

$$ffCO_2 = \frac{\delta APO - \delta APO_{bg}}{R_{\delta APO:CO_2}} \tag{12}$$

where $APO_{bg}$ is a background APO estimate, and $R_{\delta APO:CO_2}$ is the APO:ffCO$_2$ ratio which can be estimated from $R_{APO:CO_2} = \alpha_f - \alpha_B$.





To estimate the time-varying ratio $R_{\delta APO:CO_2}$ in the air intercepted at the measurement site, we use the footprint-weighted fossil fuel exchange ratio:

$$R_{t,\delta APO:CO_2} = \frac{1}{\sum_{j=0}^{n} fp_{t,j}} \sum_{j=0}^{n} (\alpha_{Ft,j} - \alpha_B) fp_{t,j} \tag{13}$$

where $t$ is the time, $j$ is the grid cell and $n$ is the maximum number of grid cells, $\alpha_{Ft,j}$ is $\alpha_F$ for one grid cell at that time, $fp_{t,j}$ is one grid cell of the hourly footprint at that time, and $\sum_{j=0}^{n} fp_{t,j}$ is the sum of the footprint across all grid cells at that time.

Here we investigate how well we can retrieve ffCO$_2$ mole fraction contributions from our APO models and we also estimate ffCO$_2$ from our observation using Equation 12. These estimates are directly compared to modeled ffCO$_2$ by multiplying the NAEI–within–EDGAR flux by NAME footprints, as described in Section 3.1. Equation 12 requires an estimate of the APO

background, $\delta APO_{bg}$. When deriving ffCO$_2$ from the model we compare two methods to estimate this term: in one case by fitting a baseline to the APO model using the REBS statistical fitting routine; for comparison we use the adjusted JC background estimate. The baselines for the whole of 2015 are shown in Supplementary Figure S9. We then derive ffCO$_2$ from the below-baseline APO, comparing the effect of using of a constant value for $R_{\delta APO:CO_2}$ and that using Equation 13 to calculate a time varying exchange ratio.

## 3 Results and discussion

### 3.1 Simulated APO at UK measurement sites

Here we show our APO model results for 2015. As examples, one summer (August) and one winter month (December) are shown throughout, and simulations for all months of 2015 and 2021 are provided in the Supplement (Figures S3 and S6).

The simulated CO$_2$ and O$_2$ mole fraction and APO contribution due to each source and sink is shown in Figure 4 for August

and December 2015 at the three sites. In August, the ocean and fossil fuel mole fraction contributions have similar magnitudes and there are sustained periods during which the ocean APO component dominates over the fossil fuel. We find that there are O$_2$ excursions from background which are considerably larger than those inferred by Pickers et al. (2017). However, there is large disagreement between the three models of ocean APO contribution, and frequently the difference between them is of a similar magnitude to that of their contribution. Whereas over the summer the ED and JC models suggest net oxygen release

from the ocean, over the winter we see overall uptake due to the difference in temperature and solubility, as well as the balance of respiration and productivity. In December, the magnitude of the fossil CO$_2$ and O$_2$ mole fractions are significantly larger than that of the ocean, although there are still large differences between the ocean models. However, when converted to the fossil fuel and ocean components of APO, the magnitudes are similar for Weybourne and for much of December the fossil fuel component is small compared with the ocean at Heathfield and Ridge Hill, despite these sites being further inland than

WAO. For all three sites, variation between the ocean models is comparable to the magnitude of their flux and there are large periods of December during which the ocean is dominant as an O$_2$ sink. This is in contrast to the findings of Chevalier and







**Figure 4.** The regional contribution of the ocean and fossil fuel components of APO to the mole fraction of each species at Weybourne, Heathfield, and Ridge Hill (*panels a, b, and c*) and the overall regional ocean and fossil fuel contribution to the APO model at the three sites (*panels d, e, and f*) throughout August 2015. The blue, green, and purple line show the contribution calculated from the ED, JC, and NE fluxes respectively, and the orange lines show the fossil fuel contributions. Solid lines represent $O_2$ in the top panels and APO in the bottom panels, dashed lines show the $CO_2$, and dash-dotted lines show the $N_2$.







**Figure 4.** continued: the regional contribution of the ocean and fossil fuel components of APO to the mole fraction of each species at Weybourne, Heathfield, and Ridge Hill (*panels g, h, and, i*) and the overall regional ocean and land contribution to the APO model at the three sites (*panels j, k, and l*) throughout December 2015. The blue, green, and purple line show the contribution calculated from the ED, JC, and NE fluxes respectively, and the orange line show the fossil contributions. Solid lines represent $O_2$ in the top panels and APO in the bottom panels, dashed lines show the $CO_2$, and dash-dotted lines show the $N_2$.



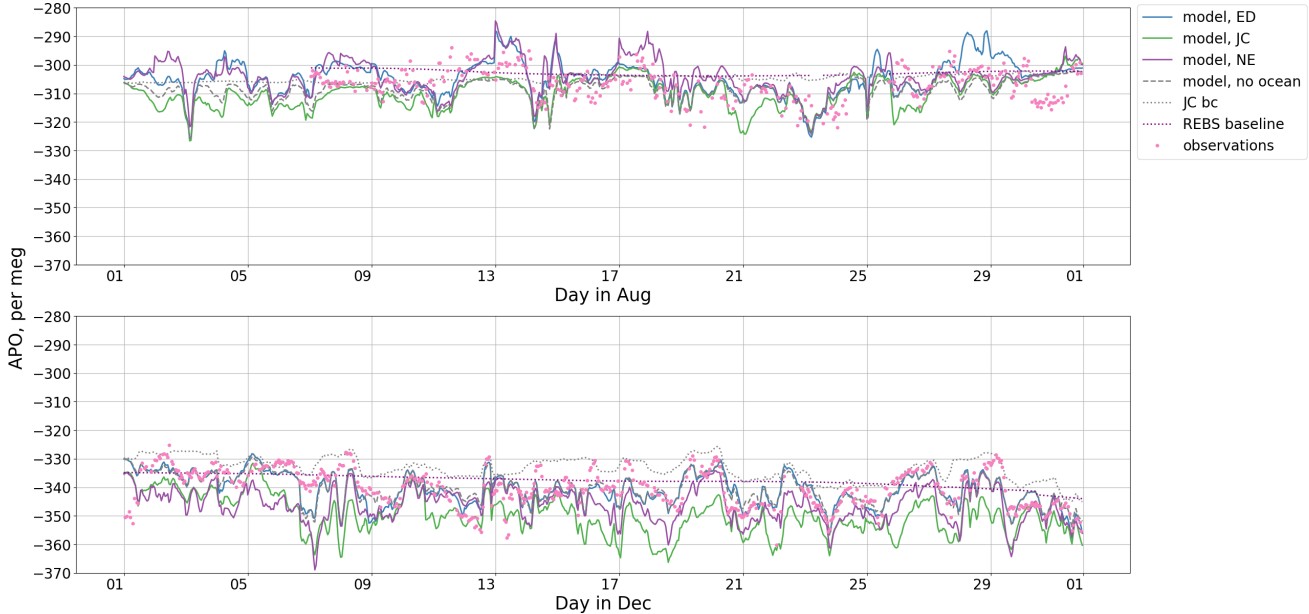

**Figure 5.** The modelled and observed APO at Weybourne throughout August (*panel a*) and December (*panel b*) 2015, where we model APO using three different ocean flux estimates from: the global ED ocean model (blue), the global JC inversion (green), and the regional NE ocean model (purple). We also show the APO model with no ocean contribution (grey dashed line). The dotted grey line shows the baseline derived from JC boundary conditions, which has been adjusted as described in Section 2.2.3. The magenta dots show the observations and the purple dotted line shows the baseline fit to the observations using the statistical fitting routine REBS.

WP4 CHE partners (2021), who found that the fossil fuel APO contribution was dominant at all sites, including Weybourne and Heathfield. That study used a combination of fluxes from NEMO – PlankTOM5 and the atmospheric transport model STILT (Lin et al., 2003). However, Chevalier and WP4 CHE partners (2021) do not provide details on the magnitude of variability in

these flux estimates.

Combining the APO components using Equation 5 gives a modelled APO for Weybourne as shown in Figure 5 (for all three sites in 2015 see Supplementary Figure S3, and for Weybourne and Heathfield in 2021 see Supplementary Figure S6). Comparing with the observations we find that, although the magnitude of the variability is similar, there are substantial differences between the simulations and the observations. Figure 6 shows the ($R^2$) and root mean squared error (RMSE), comparing each

model and the observations at Weybourne for each month throughout 2015. The APO model for December gives a closer fit to the observations at Weybourne than the model in August (average $R^2$ of 0.24 vs 0.10 and average RMSE of 6.7 vs 9.9 per meg for December and August, respectively). We see a clear seasonal trend, that the correlation is lowest throughout the summer and winter and increased during the spring and autumn. This is demonstrated further in Supplementary Figure S4, where there is larger scatter over the summer months. As discussed above and shown in Figure 5, we also find that the model

is more sensitive to the ocean flux over the summer, when the difference between the three APO simulations using different





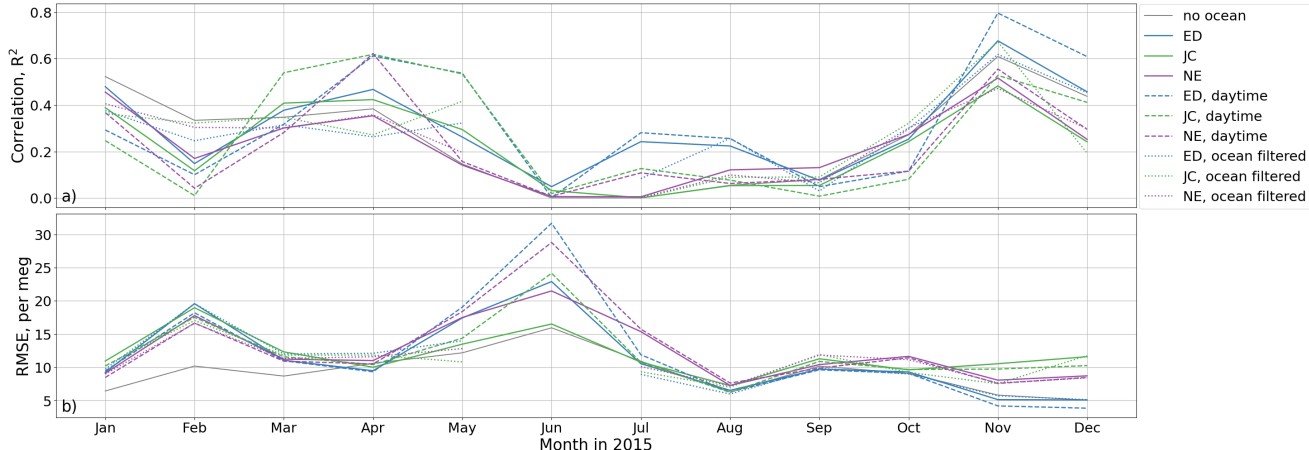

**Figure 6.** $R^2$ (*panel a*) and the root mean squared error (RMSE, *panel b*) of the modelled APO, compared with the observations at Weybourne in 2015. The blue, green, purple, and grey lines show the results from the models derived using the NAME simulations and either ED, JC, NE, and or no ocean fluxes, respectively. The solid, dashed, and dotted lines respectively show the correlations when we do not apply any filter, and when we filter for just daytime hours, and for times when the footprint has at least 40 % sensitivity to the land.

ocean fluxes is substantially larger (a monthly average of 7.0 per meg difference between the smallest and largest estimate in August, compared with 3.8 per meg in December). However, although our model agreement may be affected by ocean fluxes, we do not see a substantially better or worse fit when we exclude the ocean fluxes entirely, as shown in Figure 6. The $R^2$ and RMSE for the $CO_2$ and $O_2$ models are shown in Figure S5 of the Supplement, where we generally see higher correlations with
the data for the $CO_2$ and $O_2$ simulations ($R^2$ generally above 0.4) than we do for APO. We also find that our 2021 model, shown in Figure S6 in the Supplement, does not display such large variability. In that simulation, we use ocean climatologies, finding that localised ocean emission or uptake events are smoothed as they are averaged across a number of years.

     Next we try filtering our model in two ways to see the effects on the correlation with the observations. First we study only daytime hours (between 11:00 and 15:00), as the boundary layer is generally more well-mixed during the day than at night and
so it is often assumed that the model-data mismatch will be smaller. Separately, we filter for times at which the footprint has at least 40 % sensitivity to the land, to investigate the effects of reducing the influence of ocean-dominated time steps. With both tests we see a small improvement in the correlation in some months, although overall, the difference with the simulations with no filtering is small (Figure 6). We further discuss the sensitivity to the ocean fluxes in Section 3.4.

## 3.2 Sensitivity to exchange ratios

The 3-$\sigma$ sensitivity of APO to $\alpha_B$ and $\alpha_F$ is shown in the top and bottom panels of Figure 7, respectively (3-$\sigma$ is shown so that changes can be readily seen). In general, the model is more sensitive to $\alpha_F$ than $\alpha_B$ (average 1-$\sigma$ interval of 0.27 and 0.41 per meg for $\alpha_B$ in August and December 2015 respectively, compared to 0.30 and 0.52 per meg for $\alpha_F$). For both variables,





**Figure 7.** The APO at Weybourne during August (*panels a and b*) and December 2015 (*panels c and d*) and the sensitivity to $\alpha_B$ and $\alpha_F$. The magenta points are the observations, the purple line is the model using NE ocean $O_2$ fluxes, and the shaded region is the three $\sigma$ range derived from a Monte Carlo ensemble in which $\alpha_B$ (purple, *panels a and c*) and $\alpha_F$ (grey, *panels b and d*) are sampled.





| | August 2015 | | | December 2015 | | |
|---|---|---|---|---|---|---|
| | NAEI | EDGAR | NAEI-GridFED | NAEI | EDGAR | NAEI-GridFED |
| NAEI | | 0.957 | 0.999 | | 0.910 | 0.994 |
| EDGAR | 0.957 | | 0.962 | 0.910 | | 0.911 |
| NAEI-GridFED | 0.999 | 0.962 | | 0.994 | 0.911 | |

**Table 1.** $R^2$ for August and December 2015, comparing the modelled APO using NAEI $CO_2$ fluxes and exchange ratios, EDGAR $CO_2$ fluxes with GridFED exchange ratios, and NAEI $CO_2$ fluxes with GridFED exchange ratios. For these APO models we use the NE $O_2$ ocean flux estimates.

the influence on APO of a 1-$\sigma$ change is generally small compared with the difference between the observations and the model that we see in Figure 5. We see larger sensitivity to both values of $\alpha$ when the mole fraction is dominated by fossil fuel fluxes.

### 3.3 Sensitivity to fossil fuel $CO_2$ flux

Figure 8 shows APO at Weybourne, with fossil fuel sources modelled using a combination of fluxes and exchange ratios as follows: NAEI (within EDGAR) with NAEI exchange ratios (labeled "NAEI"), EDGAR with GridFED exchange ratios ("EDGAR-GridFED"), and NAEI with GridFED exchange ratios ("NAEI-GridFED"). We find that, although there are variations in the magnitude at some time steps, the variability of the EDGAR and NAEI fossil fuel APO models is very similar. For the most part, the two models agree, with high $R^2$ in both August and December 2015, as shown in Table 1. This suggests that the choice of inventory does not have a significant impact on the simulations compared with the other components that we investigate. Additionally, in agreement with the findings of section 3.2, the model does not seem highly sensitive to $\alpha_F$: the application of different fossil fuel exchange ratios to estimate the $O_2$ uptake does not cause strong disagreement between the two fossil fuel $O_2$ models in Figure 8, which have a high $R^2$.

Figure 8 shows the modelled APO timeseries and the associated 3-$\sigma$ range when sampling fossil fuel emissions magnitude with a 10% standard deviation. The sensitivity is highest when the air comes from populated areas. However, these periods of high sensitivity do not necessarily coincide with times when the discrepancy between the model and observations is highest, suggesting that errors in fossil fuel fluxes alone could not explain some of the differences between the model and observations.

### 3.4 Sensitivity to ocean flux

When comparing APO models and observations in Figure 6 (and Figures S3 and S4 of the Supplement), we find the biggest disagreement during the summer. At this time of year there is increased ocean productivity compared to over the winter, thus the variations between the models are larger and the APO models vary more widely. Conversely, the highest correlation between all models and the observations is seen in October (see Figure S7 of the Supplement), when the ocean acts as a small $O_2$ sink, and the $O_2$ ocean flux is smallest of any month. We see in Figures 4, 5, and Supplementary Figure S3 that the models using the ED and NE fluxes exhibit large events of $O_2$ release throughout the summer, which are more exaggerated in NE. At some



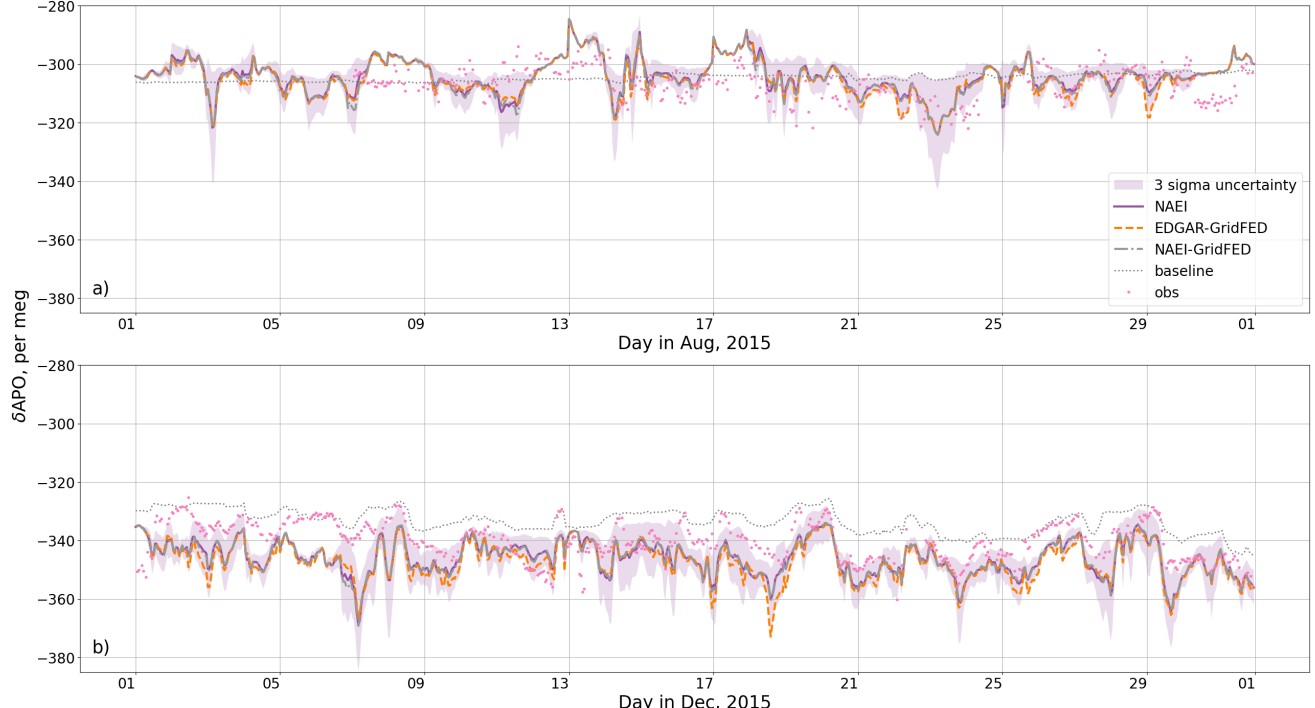

**Figure 8.** The APO model at Weybourne in August (*panel a*) and December (*panel b*) 2015 using NAME footprints and $O_2$ fluxes from the NE ocean model, comparing the model using with NAEI fluxes and exchange ratios (purple), with that using NAEI fluxes and GridFED exchange ratios (grey), and that using EDGAR fluxes and GridFED exchange ratios (orange). The observations are shown in magenta, the shaded regions represent the $3\sigma$ uncertainty in the model assuming a 10 per cent $1\sigma$ uncertainty on the fossil fuel component, and the grey dotted line is the background derived from JC boundary conditions.

of these times we see large differences between the ED and NE models compared with the model with no ocean component, as the ocean models indicate large APO excursions. Between April and June especially there are excursions in the NE APO model which have a much larger magnitude (up to ∼85 per meg) than any in the observations. On the other hand, JC shows much smaller $O_2$ fluxes with generally smoother variations, and even suggests some negative APO contribution from the ocean
during the summer. At some points during the summer we therefore see increased variability with NE compared with the other models. This difference may be due to the handling of coastal fluxes and the influence of rivers, which are more finely resolved in NE with its higher spatial resolution (∼7km vs ∼18 km), and explicit nutrient input from rivers, and by a more detailed representation of phytoplankton physiological processes (e.g. variable stoichiometry). Another factor that could contribute to the differences between the estimates of $O_2$ air-sea fluxes between the ocean models is the differences in the wind products
used to drive the air-sea exchange and their spatial and temporal resolution.



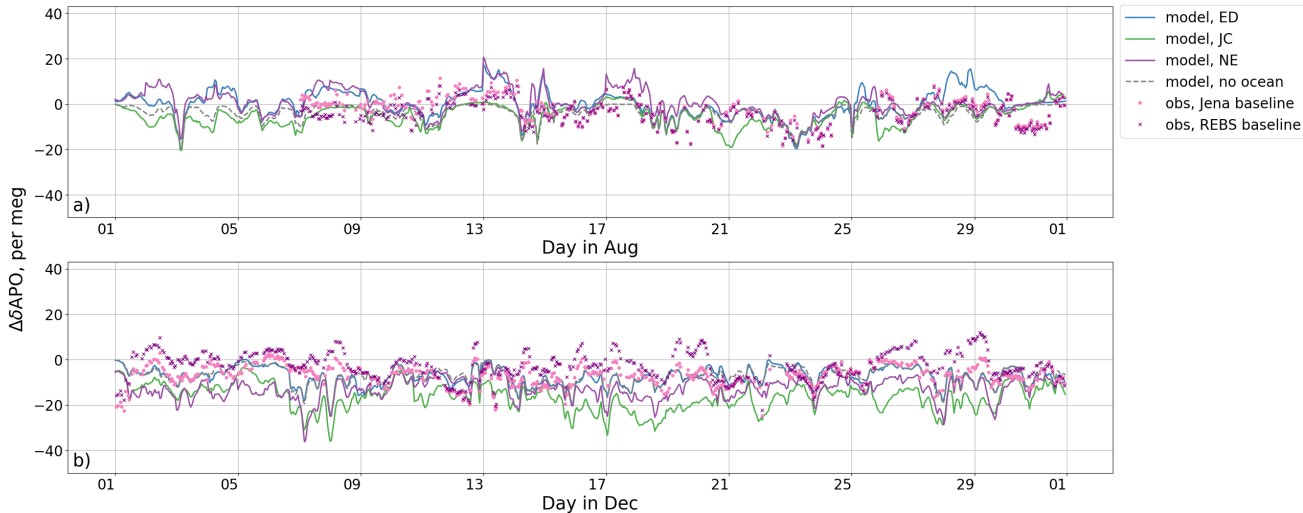

**Figure 9.** The modelled regional APO contribution and the background-subtracted APO observations at Weybourne throughout August (*panel a*) and December (*panel b*) 2015, where we model APO using three different ocean flux estimates from: the global ED ocean model (blue), the global JC inversion (green), and the regional NE ocean model (purple). We also show the APO model with no ocean contribution (grey line). We show two versions of background subtraction using a statistical routine (REBS, purple crosses), and using the JC background (pink points).

Based on our investigation we cannot determine which, if any, of the ocean flux estimates best represent the APO contribution to sites in the UK on average, although there may be some events during the summer in the NE and ED simulations that are inconsistent with the data. Furthermore, we do not see a substantial difference in correlation between the observations and either the simulations that include ocean fluxes or that which does not.

## 3.5 Sensitivity to the background estimate

Figure 9 shows the modelled regional $\Delta(\delta\text{APO})$ and the background-subtracted observations. We compare the background subtraction from the statistical (REBS) filter with the adjusted model-estimated baseline from the JC global fields. For most of the time series, the two baseline estimates lead to similar regional signals. In December there is more of a difference between the two signals, where the at some regions the REBS subtracts a smaller background and leaves positive APO excursions. We expect that this difference arises because there is more variability within the JC background estimate. We saw in Figure S2 of the Supplement that this variability is increased in the winter compared to summer. We see in Figure 10 that the correlation between the background-subtracted observations and the models is similar for both methods of background subtraction. Neither choice leads to a substantial difference in model-data mismatch.





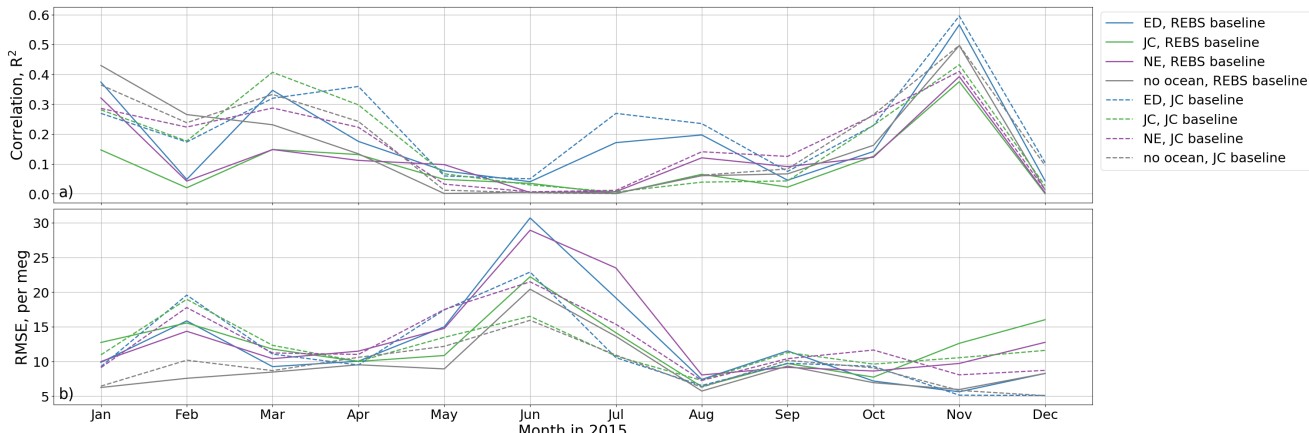

**Figure 10.** The square of the Pearson correlation coefficient ($R^2$, *panel a*) and the RMSE (*panel b*) of the modelled regional contribution of APO, compared with the background subtracted observations at Weybourne in 2015. The blue, green, purple, and grey lines show the results from the models derived using the NAME simulations and either ED, JC, NE, or no ocean fluxes respectively. The solid and dashed lines respectively show the results when we subtract the REBS statistical background from the observations, and when we subtract the JC derived background.

### 3.6 Estimation of fossil fuel $CO_2$

Here we test how well we can retrieve the regional contribution of $ffCO_2$ from our modelled APO, using the method described in Section 2.4. Figure 11 and Supplementary Figure S10 show the comparison between $ffCO_2$ derived from our modelled APO and the direct simulation of $ffCO_2$ using NAME (i.e., $ffCO_2$ fluxes multiplied by NAME footprints). The comparison for all months throughout 2015 and the correlations are shown in Supplement Figure S10. Comparisons are shown when three different ocean flux estimates are used, or two different methods for subtracting the baseline. Differences between the APO-derived $ffCO_2$ and the direct $ffCO_2$ simulation will be due to the influence of ocean fluxes on the APO simulation (which is assumed negligible in Equation 12) and mis-specification of the background. All other factors, including atmospheric transport, are consistent between the two sets of simulations. Therefore, the APO-derived $ffCO_2$ using the adjusted JC background exactly matches the direct $ffCO_2$ simulation, if ocean fluxes are zero.

Firstly, we will consider the APO-derived $ffCO_2$ using the adjusted JC backgrounds. Throughout the summer, when there are large $O_2$ release events in the modelled ocean fluxes, the APO simulation using NE generally underestimates $ffCO_2$, even indicating negative mole fractions for large parts of the month. The ED and JC APO simulations show closer overall agreement with $ffCO_2$ in August, although some discrepancy remains for all three.All three models overestimate the $ffCO_2$ for the majority of the winter compared to the direct $ffCO_2$ simulation. In this case the background APO, estimated as described in Section 2.4, is underestimated for large parts of the month, which may be due to modelled oceanic uptake of oxygen around the UK throughout the winter. Chevalier and WP4 CHE partners (2021) found high correlations between their APO-derived $ffCO_2$





**Figure 11.** The modelled ffCO$_2$ for August (*panels a, b, and c*) and December (*panels d, e, and f*) 2015 derived from the APO model for Weybourne using the results from three different ocean flux fields (blue): ED (*panels a and d*), JC (*panels b and e*), and NE (*panels c and f*). We compare with the model calculated directly from the NAEI-within-EDGAR fluxes and NAME footprints (pink). The direct model is equivalent to the ffCO$_2$ in the top panels of Figure 4 and the APO models are shown in Figure 5.





and direct STILT model. However, it is unclear from that work as to the time period over which this correlation was found, and it should be noted that our correlation is greatly improved when averaging over larger time periods, due to the seasonality in APO.

For the simulations in which the REBS baseline has been fit to the APO simulations and then subtracted, the derived ffCO$_2$
from ED and NE is higher during the summer and lower during the winter than when the adjusted JC background is used. For the model that used JC ocean fluxes, which are considerably smaller than either ED or NE, there is a much smaller difference between the two estimates. The large difference between the simulations using these two baseline estimates likely stems from the influence of ocean fluxes. The REBS fit incorporates seasonal oceanic trends and thus removes large-timescale oceanic fluxes from the model. However, it is also susceptible to fitting to large APO excursions in the model which occur due to
modelled short-term variability from the ocean, this is particularly clear throughout June in Figure S9 of the Supplement. On the other hand, as JC is independent of the model it does not encapsulate any regional ocean influence, and any ocean contribution is treated as ffCO$_2$.

In Section 2.4 we make the assumption that the ocean component of the APO measurements is negligible when deriving ffCO$_2$. This is based on previous studies of short-term ocean-related APO variability, which in turn are based on observations.
Yet these models all indicate a persistent ocean contribution at all sites, which biases our calculation of ffCO$_2$ from the APO simulations. As shown in Section 3.1, there is large variation in O$_2$ flux estimates between ocean models. However, we cannot conclude which model, if any, gives a more accurate representation of the ocean O$_2$ flux. Furthermore, the CO$_2$ and O$_2$ ocean fluxes are decoupled and therefore, the exchange ratio varies as the footprint intercepts different parts of the ocean. Based on our analysis using these three ocean flux estimates, a correction for oceanic fluxes would be subject to substantial uncertainty.

Next we apply the same method to estimate ffCO$_2$ from the observed APO at Weybourne (Pickers et al., 2022) as described in Section 2.4. Figure 12 shows observation-derived ffCO$_2$ compared with the direct ffCO$_2$ simulations. Here, we have used the NAME simulation with NAEI and EDGAR fluxes, and also the outputs of the CTE system. The correlations (R$^2$) between the observation-derived ffCO$_2$ and the ffCO$_2$ model are shown in Figure 13. As we found in Section 2.2, we generally see low correlations over the summer, with stronger agreement in March, April, and November. There is not a large difference in the
correlation for the JC and REBS background subtractions. This is contrary to our findings above shown in Figure 11, where we saw that there was sometimes large differences in ffCO$_2$ estimates for different methods of background subtraction due to the large ocean contribution which was assumed to be encapsulated in the background estimate. Throughout December we see that when using the REBS background subtraction we estimate frequent negative ffCO$_2$ contributions, which are not as apparent when subtracting the JC background, which may be a result of increased variability of the JC background estimate.
Based on the synthetic data results presented in the previous paragraphs, discrepancies may be because of the influence of non-negligible ocean flux contributions, or errors in assigning baseline values. At some times we see a $\sim 5-8\,\mu$mol/mol difference between the direct model and the observation-derived ffCO$_2$ using the REBS background subtraction; this translates to an ocean contribution of $\sim 10-20$ per meg. This would be a large contribution, although the majority of the differences between the estimates are much smaller than this.





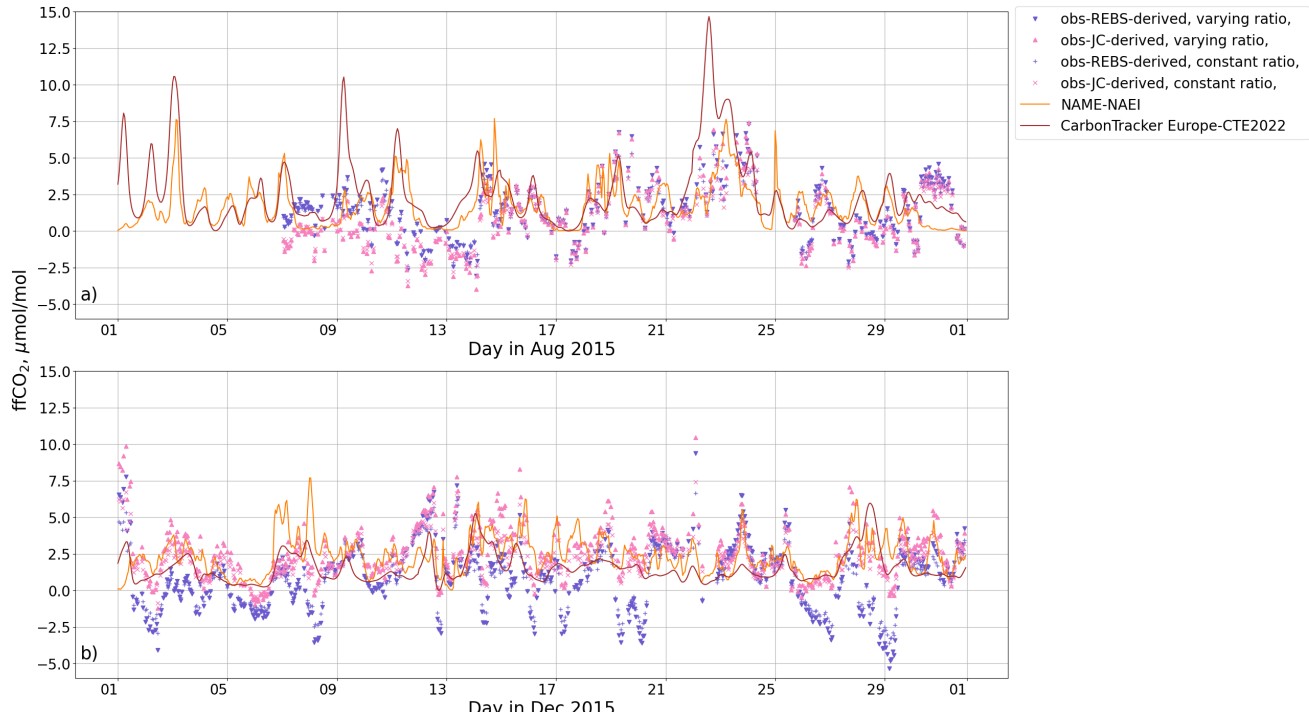

**Figure 12.** The regional contribution of ffCO$_2$ to the atmospheric abundance at Weybourne for August (*panel a*) and December (*panel b*) 2015. The pink triangles and crosses show the ffCO$_2$ model derived from the APO observations with the JC background subtracted using a time-varying and a constant exchange ratio respectively, the purple triangles and pluses show the same but with the REBS baseline subtracted, the orange line shows the model calculated directly from the NAEI-within-EDGAR fluxes and NAME footprints (equivalent to that in the top panels of Figure 4) and the brown line shows the model derived from CarbonTracker Europe (CTE2022).

We also test the conversion of the APO observations to ffCO$_2$ using a constant APO:ffCO$_2$ ratio, assuming $\alpha_F = -1.5$, as shown by the blue points in Figure 12. Throughout the year, the correlation between this estimate of ffCO$_2$ and the direct model are slightly lower than when using a time-varying APO:ffCO$_2$ ratio. Thus we find that using a time-varying APO:ffCO$_2$ ratio gives a slightly closer fit to the direct ffCO$_2$ simulation.

## 4    Future outlook

Improvements in the measuring and modelling of tracers are important for future evaluation of ffCO$_2$ emissions. Our investigation has shown that there are several inputs which can sometimes substantially change the modelled APO. In particular, a better understanding of oceanic CO$_2$ and O$_2$ fluxes in coastal regions seems to be the most important of the factors we tested, as such continental sites far from ocean influence may currently be more viable for APO models. We also saw that the choice





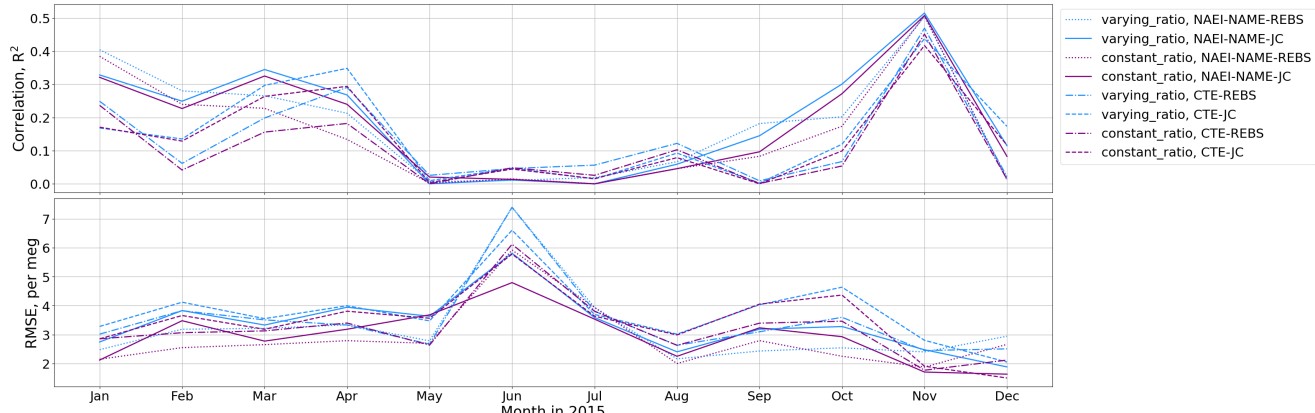

**Figure 13.** The square of the Pearson correlation coefficient ($R^2$, *panel a*) and the root mean squared error (RMSE, *panel b*) of the modelled APO, compared with the observations at Weybourne in 2015. The blue and purple lines show the correlation when using a time-varying and constant APO:ffCO$_2$ ratio respectively, the solid lines show the correlation between the NAME-NAEI model and the JC-background-subtracted observations, and the dotted lines show the same but with the REBS-background-subtracted observations. The dashed lines show the correlation between the CTE model and the JC-background-subtracted observations, and the dash-dotted lines show the same but with the REBS-background-subtracted observations.

of baseline affects our APO model and derived ffCO$_2$, although errors in assigning regional baselines may also be due in part
to the influence of non-terrestrial fluxes.

Alongside APO, other tracers such as radiocarbon and CO can give extra insight into ffCO$_2$ emissions. Several studies have shown that radiocarbon is a promising tool for this (e.g. Levin et al., 2003; Graven et al., 2009). However, unlike APO, most radiocarbon programs rely on flask measurements which are not continuous and require time-consuming analysis. This makes radiocarbon a comparatively expensive method which cannot presently provide such insight into high-frequency variability.
Radiocarbon measurements are also susceptible to contamination of emissions from the nuclear power industry, correcting for which requires access to data which is not currently publicly available in the UK. Although CO measurement are much cheaper than radiocarbon and can be made continuously (e.g. Andrews et al., 2014; Levin and Karstens, 2007; Levin et al., 2020), the conversion from CO to ffCO$_2$ is uncertain.

Given the challenges of each, no one tracer currently provides the answer to the verification of ffCO$_2$ emissions. However,
we have identified key areas of focus which may improve the modelling of APO and its use as a ffCO$_2$ tracer.

## 5   Conclusions

We have simulated the tracer APO throughout the years 2015 and 2021 at three sites in the UK: Weybourne, Heathfield, and Ridge Hill. Generally, the correlation with the observations is small for APO. We find large modelled ocean signals which



sometimes dominate the APO model, and that correlations tend to be higher for APO during the spring and autumn when
ocean fluxes are smallest.

We have presented a sensitivity analysis of the factors that most strongly influence modelled atmospheric APO. Our simulations suggest that uncertainties in ocean fluxes contribute substantially to modelled APO and APO-derived ffCO$_2$ from the model at measurement sites in the UK. Our analysis cannot determine which ocean model (or indeed, zero ocean flux) or baseline estimation method leads to closest agreement the observations. However, a robust estimate of ffCO$_2$ is likely to depend
strongly on these factors being well known. In comparison, the sensitivity of atmospheric APO to uncertainties in fossil fuel and terrestrial biosphere exchange ratios was relatively small. Our analysis shows that further work should focus on improving ocean O$_2$ and CO$_2$ flux estimates to improve the accuracy of high-frequency APO-derived ffCO$_2$ in the UK.

## 6 Code Availability

The code for the analysis presented is available at https://github.com/hanchawn/APO_modelling (Chawner, 2023). We also
use code developed by the ACRG Modelling team at the University of Bristol, which is available at https://github.com/ACRG-Bristol/acrg.

## 7 Data Availability

The datasets generated and analysed during this study are available at https://zenodo.org/record/7681834 (Chawner et al., 2023). The observational datasets are available on CEDA at:

– Heathfield CO$_2$ and O$_2$: https://catalogue.ceda.ac.uk/uuid/bfc2483537a744dca8e3239278b6e522

    – Weybourne CO$_2$: https://catalogue.ceda.ac.uk/uuid/87fc265aab6b4aeb961e62da2cd6ca91

    – Weybourne O$_2$: https://catalogue.ceda.ac.uk/uuid/b3f9714c956f428a840211e0184e23eb

## 8 Author contribution

HC carried out the atmospheric modelling and data analysis with contributions from PAP, YA, AJM, CR, GL, PL and ITL.
Measurements were made by KEA and PAP, with support from TA, CD, GLF and CR. HC wrote the paper, with contributions from MR, PAP, YA, ITL, HG, ALG and all co-authors.

## 9 Competing interests

The authors declare that they have no conflict of interest.



## 10 Acknowledgements

HC, MR and ALG were supported by a Natural Environment Research Council grant to the University of Bristol as part of the Detection and Attribution of Regional Emissions (DARE-UK) project, NE/S004211/1. We thank P. Wilson, and T. Barningham for assisting with maintaining the WAO $O_2$ and $CO_2$ measurement system during 2015.

Atmospheric $O_2$ and $CO_2$ measurements at WAO in 2015 and 2021 were funded by the U.K. Natural Environment Research Council (NERC) grants NE/I013342/1, NE/S004521/1, and NE/R011532/1. The WAO atmospheric $O_2$ and $CO_2$ measurements

have also been supported by the U.K. National Centre for Atmospheric Science (NCAS) from 1st December 2013 onward. P.A.P., K.E.A., and G.L.F. received funding from the NERC project DARE-UK (NE/S004211/1), and P.A.P and K.E.A. have received funding from the Horizon Europe project PARIS (101081430).

YA and GL acknowledge DARE-UK (NE/S004947/1) and the U.K. National Capability NERC Climate Linked Atlantic Sector Science program (NERC grant no. NE/R015953/1).



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
