# Peer review of "Atmospheric oxygen as a tracer for fossil fuel carbon dioxide: a sensitivity study in the UK"

_EGUsphere, 2023_

## Referee Comment (RC1)

Overall, this is a clear and thorough examination of the use of APO as a tool for quantifying fossil fuel CO2 emissions. The authors have conducted numerous sensitivity studies and model comparisons and conclude that APO can't give robust estimates of fossil CO2 emissions with our current knowledge of oceanic APO fluxes.

The results here are important, the work is sound and the presentation is good. I have some corrections and suggestions to offer, but most of these are minor and I feel that the paper should definitely be published (after my concerns are addressed).
* * *
**Three substantive matters come to mind that are worthy of attention:**

This work relies heavily on the Jena Carboscope (JC) Inversion for background values and also regional oceanic fluxes. I am little concerned that there is no mention of the risk of circularity here: JC is an inversion that is based in part on the data from Weybourne. I would like to be convinced that the JC results give a background estimate that is truly independent of the record from which it is being subtracted.

The abstract and introduction led me to believe that all three sites (Weybourne, Heathfield & Ridge Hill) would play significant roles in this work. While Heathfield and Ridge Hill do add substance and breadth to the analysis, the truth is that Weybourne is by far the most important site in this study and most references to the other two are relegated to the supplement. I would prefer that the secondary nature of Heathfield and Ridge Hill was more explicitly acknowledged throughout. Anything else feels like bait-and-switch tactics.

Finally, the "future work" section is limited to developing better/more certain ocean fluxes. This ignores two possible avenues for exploration: a different species that might give insight into oceanic oxygen fluxes (noble gases?), or a different oxygen-based "tracer". APO was originally formulated to eliminate terrestrial signals and be as sensitive as possible to oceanic oxygen fluxes. Any ability to learn about fossil CO2 fluxes from APO is simply luck, capitalizing on the fact that $\alpha_F \neq \alpha_B$. However, these oxidative ratios aren't actually very different. I am left wondering whether one would be better able to use oxygen for quantifying fossil CO2 fluxes by combining O2 and CO2 into a tracer that was minimally sensitive to oceanic fluxes and then use terrestrial models (which might be better constrained than oceanic models, in this context) to take out the land contribution, leaving a more robust fossil signal. I recognize that actually doing this work is well beyond the scope of this study, but a brief mention of these alternative approaches (or others) would make the "future work" section much less *pro forma*.

**A few minor requests for elaboration or clarification:**

Line 56: I would like to see a sentence comparing this work to the aforementioned studies of Kuijpers and CHE. In particular, state your expectations for the UK sites. Do you expect to see just what the other studies saw, or do you expect to be more sensitive to, for example, marine influences?

Line 213-214: If this is a period of minimal terrestrial influence, why are you comparing to a simulation with no ocean fluxes? Perhaps this is a standard modelling practice, but without more detail, I don't

understand what is meant by "the 90 percentile of APO in a simulation with no ocean fluxes", nor how it defines a period of minimal terrestrial influence.

Lines 229ff: As I understand it, you are asking "If the actual emissions of ffCO2 go up or down, will we capture those variations if we start with APO and infer the ffCO2 emissions?" However, instead you state: "we study the sensitivity of the modelled fossil fuel contribution to the atmospheric concentration of CO2 and O2". I believe you are using "modelled fossil fuel contribution" for the value of ffCO2 inferred from APO, and "the atmospheric concentration of CO2 and O2" is actually the variation in atmospheric CO2 and O2 mole fractions arising solely from fossil fuel combustion. If I am correct, I find your wording very confusing. If I am not correct, I am truly confused. Either way, please clarify.

Lines249ff: Again, only after close study do I *think* I understand what you're doing here. I believe the crucial point here is that "modelled $\Delta(\delta APO)$ (calculated using equation 5)" is a fully model-based prediction of the change in APO that results solely from sources and sinks *within the region*. If I am correct about this, please state it more explicitly.

**Very minor editorial points:**

Line 12: should read "contribution of simulated fossil fuel CO2 to APO."

Line 37: New paragraph beginning "When considering ocean fluxes…"

Lines 61 & eq 1: The spacing in "reference" is odd. Maybe this is just a quirk of latex. Did you use \mathit for the subscript?

Lines 75 and following: Please be a little more explicit about the units in these eqs. In particular, it requires some work for the reader to determine whether $\Delta(\delta APO)$ is a difference (in permeg) or a flux (in permeg/year) or some other units.

Line 85: The citation of Pickers is incomplete.

Line 87: "Variations in…" is a run-on sentence. A new one should begin at "however".

Line 144: Should read "for the influence of rapid variations in CO2 flux on the mole fractions, footprints are…"

Figure 3 caption: I assume panel D shows the footprint for WAO. This is not stated in the caption.

Figure 4: In the figure caption, the distinction between the groups of panels (a,b,c vs. d,e,f) is opaque. In my mind, there is no meaningful difference between "regional contribution" and "overall regional [contribution]". I believe you have just combined the four respective sets of traces for CO2 and O2 in a,b,c (with appropriate weighting) to get APO in d,e,f. The word "overall" does not convey this relationship. Please make this easier to discern from the caption.

Line 320: "The APO model…" Exactly which model? By eye it's not obvious which lines have been included in the average, yielding the quoted $R^2$ value of 0.24 for December.

Figure 6 caption: Is there a superfluous "and" in the 3rd line?

Line 340: The correspondence between $\alpha_F$ and $\alpha_B$ in the panels of Fig. 7 is not as stated, and it's not simple to state correctly, so just remove "the top and bottom panels of" and "respectively".

Table 1:  The extension of the vertical line (in the column headings) separating August 2015 from December 2015 is in the wrong place.

Line 413:  Should "large-timescale" really be "long-timescale"?

Figures 6 & 10:  In the first case you simply say "$R^2$" and in the second you say "The Pearson correlation coefficient $R^2$".  Either switch the order (giving a more complete introduction and subsequent abbreviation) or just stick with the simpler $R^2$ in both cases.

---

## Author Comment (AC1)

**"Atmospheric oxygen as a tracer for fossil fuel carbon dioxide: a sensitivity study in the UK" – Response to Anonymous Referee #2**

Hannah Chawner[1], Karina Adcock[2], Eric Saboya[1,7], Tim Arnold[3,4], Yuri Artioli[5], Caroline Dylag[3], Grant L. Forster[2,6], Anita Ganesan[7], Heather Graven[8], Gennadi Lessin[5], Peter Levy[9], Ingrid T. Luijkx[10], Alistair Manning[11], Penelope A. Pickers[2], Chris Rennick[3], Christian Rödenbeck[12], Matthew Rigby[1].

[1]School of Chemistry, University of Bristol, Bristol, UK
[2]Centre for Ocean and Atmospheric Sciences, School of Environmental Sciences, University of East Anglia, Norwich, UK
[3]National Physical Laboratory, Teddington, UK
[4]School of Geosciences, University of Edinburgh, Edinburgh, UK
[5]Plymouth Marine Laboratory, Plymouth, UK
[6]National Centre for Atmospheric Sciences, University of East Anglia, UK
[7]School of Geographical Sciences, University of Bristol, Bristol, UK
[8]Department of Physics, Imperial College London, London, UK
[9]Centre for Ecology and Hydrology, Edinburgh, UK
[10]Meteorology and Air Quality, Wageningen University and Research, Wageningen, the Netherlands
[11]Hadley Centre, Met Office, Exeter, UK
[12]Max Planck Institute for Biogeochemistry, Germany
* * *
Below we describe the changes that have been made to the re-submitted manuscript "Atmospheric oxygen as a tracer for fossil fuel carbon dioxide: ac sensitivity study in the UK" as recommended in the comments from the two reviewers.

Additional small changes that primarily relate to typos spotted by the authors have also been made and are listed towards the end of this document. Please note that these small additional changes made by the authors do not substantially affect the scientific findings.

Referee comments are stated in black with author responses in blue. Proposed changes to the manuscript are clearly stated throughout (usually in bold typeface).
* * *
**RC1 referee comments**

Overall, this is a clear and thorough examination of the use of APO as a tool for quantifying fossil fuel $CO_2$ emissions. The authors have conducted numerous sensitivity studies and model comparisons and conclude that APO can't give robust estimates of fossil $CO_2$ emissions with our current knowledge of oceanic APO fluxes.

The reviewer is correct that our manuscript indicates a potentially substantial but highly uncertain contribution of oceanic fluxes to the simulated APO in the UK. The oceanic influence has not been previously demonstrated in these APO observations and remains to be further investigated. As demonstrated in the manuscript, model-data $CO_2$ and $O_2$ values are better correlated than the APO model-data comparison, indicating there is a specific challenge in APO model-data comparisons at least at the Weybourne and Heathfield sites. However, we must be careful that overly broad conclusions are not drawn from our work. Indeed, the preprint by Rödenbeck et al. (2023) *ACPD*, which was published after our manuscript was submitted, suggests that, given a network of sufficient density, fossil fuel flux estimates may be less strongly influenced by ocean fluxes, particularly for continental regions. We have checked the wording in our manuscript to ensure we are not inadvertently giving the impression that conclusions could readily be extended to other regions or network densities. We now include the following reference(s) to the Rödenbeck study.

At the end of "Section 3.4 Sensitivity to ocean flux" we include (shown in bold):
*"Based on our investigation we cannot determine which, if any, of the ocean flux estimates represent the APO contribution at sites in the UK. **Furthermore, we do not see a substantial difference in correlation between***

*the observations and either the simulations that include ocean fluxes or those that do not. Chevalier et al. (2021) also noted an ocean influence in their simulations using different transport models to those used here. Our result requires further investigation since the magnitude of some of the short-term ocean variability* during the summer in NE and ED simulations is inconsistent with *what is seen in the observations at WAO. Furthermore, it needs to be determined the extent to which these findings are due to the coastal location of WAO since some shipboard measurements do not show a large sensitivity to ocean fluxes (Pickers 2016). Rödenbeck et al. (2023) suggest that a dense continental network of stations measuring APO could minimize the potential influence of oceanic fluxes, meaning that robust estimates of fossil fuel $CO_2$ fluxes could be made by using observed APO gradients within a continent.*"

And in the conclusion:
"*Our analysis cannot determine which ocean model (or indeed, zero ocean flux) or baseline estimation method leads to closest agreement with the observations. However, a robust estimate of $ffCO_2$ is likely to depend strongly on these factors being well-known,* **or proven to have little influence using observation-based methods. We do not find evidence from our three UK stations that the substantial (yet uncertain) influence of oceanic fluxes on simulated APO is reduced further inland. But since the UK is surrounded by ocean, simulated APO at continental European locations may be less affected. More robust $ffCO_2$ estimates may be possible in general if a sufficiently dense network of sites were available, which could account for fossil fuel influences jointly with that of any oceanic sources.*"

The results here are important, the work is sound and the presentation is good. I have some corrections and suggestions to offer, but most of these are minor and I feel that the paper should definitely be published (after my concerns are addressed).

**Three substantive matters come to mind that are worthy of attention:**

1. I am little concerned that there is no mention of the risk of circularity here: JC is an inversion that is based in part on the data from Weybourne. I would like to be convinced that the JC results give a background estimate that is truly independent of the record from which it is being subtracted.

The reviewer is correct that the JC APO fields, which we have used as a boundary condition, were derived in a global inversion in which Weybourne data were used. However, we are confident that any circularity will not have a significant impact on our results. This is because we only use these fields to examine the variability within each month due to APO gradients at the regional boundary (the absolute level is adjusted each month to ensure that the mean data and the model approximately agree). Since the domain boundary is on the order of ~1000km from the site, we expect that the JC gradients in that part of the atmosphere will only be minimally influenced by the inclusion of WAO.

We now address this comment in the manuscript by including the following sentence in Section 2.2.3:

"*... of APO mole fractions from the JC global APO inversion (Rödenbeck et al., 2008, version apo99X_WAO_v2021).* **Whilst the JC APO fields include data from WAO in their derivation, any circular influence on our results should be small, because the domain boundaries are far from the UK (~1000 km) and therefore, the WAO data should not strongly influence the gradients simulated there.** *These boundary conditions are ...*"

2. The abstract and introduction led me to believe all three sites (Weybourne, Heathfield & Ridge Hill) would play significant roles in this work. While Heathfield and Ridge Hill do add substance and breath to the analysis, the truth is that Weybourne is by far the most important site in this study and most references to the other two are relegated to the supplement. I would prefer that the secondary nature of Heathfield and Ridgehill was more explicitly acknowledged throughout. Anything else feels like bait-and-switch tactics.

We have included the following sentence in the abstract to underscore that the main model-data analysis was performed for one UK site with additional comparisons presented in the Supplementary Information.

*"… from the NEMO-ERSEM and ECCO-Darwin ocean models, and the Jena CarboScope APO inversion. **We focus our model-data analysis on the year 2015 as ocean fluxes were not available for later years. As APO measurements are only available for one UK site at this time, our analysis focuses on the Weybourne station. Model-data comparisons for two additional UK sites (Heathfield and Ridge Hill) in 2021, using ocean flux climatologies, are presented in the supplement.** The sensitivity of APO to fossil fuel emissions …"*

And the following edit has been made to the last paragraph of the introduction.

*"Two measurement sites equipped with high-frequency $CO_2$ and $O_2$ instruments have been established in the UK, one at Weybourne Atmospheric Observatory (WAO) in the east of England and one at Heathfield tower (HFD) in the south of England. In this paper, we perform simulations of $CO_2$ and $O_2$ **primarily** focussing on **model-data comparisons at WAO for the year 2015, with further comparisons at HFD and WAO for the year 2021 presented in the supplement** along with a third site at Ridge Hill (RGL) **telecommunications tower**. Although $O_2$ measurements are not available at RGL, it is included to examine the modelled APO further inland."*

3. Finally, the "future work" section is limited to developing better/more certain ocean fluxes. This ignores two possible avenues for exploration: a different species that might give insight into oceanic oxygen fluxes (noble gases?), or a different oxygen-based "tracer". APO was originally formulated to eliminate terrestrial signals and be as sensitive as possible to oceanic oxygen fluxes. Any ability to learn about fossil $CO_2$ fluxes from APO is simply luck, capitalizing on the fact that $\alpha_F \neq \alpha_B$. However, these oxidative ratios aren't actually very different. I am left wondering whether one would be better able to use oxygen for quantifying fossil $CO_2$ fluxes by combining $O_2$ and $CO_2$ into a tracer that was minimally sensitive to oceanic fluxes and then use terrestrial models (which might be better constrained than oceanic models, in this context) to take out the land contribution, leaving a more robust fossil signal. I recognize that actually doing this work is well beyond the scope of this study, but a brief mention of these alternative approaches (or others) would make the "future work" section much less *pro forma*.

We have reworked this section slightly to highlight that this is a modelling study, which could now pave the way to further observation-based analysis of these important datasets. We have also noted that tracers of ocean fluxes could be sought to help understand some of the issues discussed in the manuscript (although the precise nature of what might be practical or useful here is beyond our expertise).

Regarding combining $O_2$ and $CO_2$ into a tracer that is minimally sensitive to ocean fluxes, to our knowledge, this would not be possible. Unlike for land processes, where they are strongly coupled, $O_2$ and $CO_2$ fluxes from the ocean are largely decoupled, as we mention in the introduction. This partly arises because $CO_2$ dissociates to bicarbonate and carbonate ions when it dissolves in seawater, whilst gases such as $O_2$ are not very soluble in water, thus changing the balance of carbon between the ocean and atmosphere according to Henry's law. As such, the air-sea equilibrium time for each gas is very different – about 1 year for $CO_2$ and only a few weeks for $O_2$ (Keeling and Shertz, 1992). In addition, biological and solubility driven fluxes for $CO_2$ largely cancel each other in most ocean regions, whereas for $O_2$, these fluxes are reinforcing (Stephens et al., 1998).

**A few minor requests for elaboration or clarification:**

4. Line 56: I would like to see a sentence comparing this work to the studies of Kuijpers and CHE. State your expectations for the UK sites. Do you expect to see what other studies saw, or do you expect to be more sensitive to, for example, marine influences?

Whilst we agree it is important the implications of this study are compared with the existing literature, we feel this comparison would be better placed when we are discussing our results. We have included the following lines:

Section 3.2 (sensitivity to exchange ratios): *"Chevalier et al. (2021) also identified an influence on the simulated APO due to potential misspecification of $\alpha_B$."*

Section 3.4 (sensitivity to ocean flux): *"Chevalier et al. (2021) also noted an ocean influence in their simulations using different transport models to those used here."*

5.  Line 213-214: If this is a period of minimal terrestrial influence, why are you comparing to a simulation with no ocean fluxes? Perhaps this is a standard modelling practice, but without more detail, I don't understand what is meant by "the 90 percentile of APO in a simulation with no ocean fluxes", nor how it defines a period of minimal terrestrial influence.

The idea here is to estimate an "offset" that can be applied to the model simulations in each month, so that the baseline is approximately consistent with the atmospheric data. This is necessary because we only have a regional model, so need to account for the regional background, and we don't want to be reliant on hard-wiring background levels from another inversion product (see response to Point 1). For long-lived trace gases, we often use regional model simulations to estimate data points that are minimally influenced by sources of the gas of interest (which are typically land-based) and infer the baseline from measurements made at those times (e.g., Manning et al., 2021, 2011). Here, we've attempted to do the same for APO by running a simulation with no ocean flux, and then finding the data points with the smallest deviation from zero (i.e., the highest APO values, since the deviations are negative).

The problem with this approach is that it relies on these points being close to the regional background in the observations. However, as the paper shows, for APO at WAO, even when the influence of land-based sources is minimised, there may be a deviation from the baseline because of ocean fluxes. Because of these potential issues, we investigate the influence of using different baseline estimation methods in Section 3.5.

6.  Lines 229ff: As I understand it, you are asking "If the actual emissions of $ffCO_2$ go up or down, will we capture those variations if we start with APO and infer the $ffCO_2$ emissions?". However, instead you state: "we study the sensitivity of the modelled fossil fuel contribution to the atmospheric concentration of $CO_2$ and $O_2$". I believe you are using "modelled fossil fuel contribution" for the value of $ffCO_2$ inferred from APO, and "the atmospheric concentration of $CO_2$ and $O_2$" is actually the variation in atmospheric $CO_2$ and $O_2$ mole fractions arising solely from fossil fuel combustion. If I am correct, I find your wording very confusing. If I am not correct, I am truly confused. Either way, please clarify.

The reviewer is correct that the wording is confusing here. We have modified this section to read:

*"We estimate the sensitivity of the modelled APO to changes in the distribution and magnitude of fossil fuel $CO_2$. We investigate the influence of the spatial distribution by comparing APO simulations for the NAEI and EDGAR, which are overall very similar in magnitude, but have a different distribution (Figure 2). As discussed in Section 2.2, our APO model uses NAEI $ffCO_2$ emissions estimates for the UK, which are embedded in those of EDGAR and combined with NAEI fuel usage statistics to calculate $ffO_2$ uptake. We compare these estimates to EDGAR $CO_2$ emissions with GridFED αF.*

*We investigate the sensitivity of the APO model to the magnitude of $ffCO_2$ using a Monte Carlo ensemble in which the overall $CO_2$ flux in the entire domain is allowed to vary by ±10%. This range is considerably larger than the difference between EDGAR and the NAEI, which is approximately 0.7%, but chosen so that the effect on APO can be readily identified."*

7.  Lines 249ff: Again, only after close study do I *think* I understand what you're doing here. I believe the crucial point here is that "modelled $\Delta(\delta APO)$ (Calculated using equation 5)" is a fully-modelled based prediction of the change in APO that results solely from sources and sinks *within the region*. If I am correct about this, please state it more explicitly.

This is correct. We have reworded the first few lines of this subsection to make this clearer:

*"As our APO simulations only account for the influence of fluxes within our regional domain, an estimate must be made of the APO entering the domain. Therefore, in this section, we describe how different background estimates might influence the comparison between the APO simulation and the observations. The background represents the APO variability that is representative of the well–mixed atmosphere at the UK's latitude, excluding local influences. We compare the modelled $\Delta(\delta APO)$ …"*

**Very minor editorial points**

8.  Line 12: should read "contribution of simulated fossil fuel $CO_2$ to APO"

This has now been corrected in the manuscript.

9.  Line 37: New paragraph beginning "When considering ocean fluxes …"

This has now been corrected in the manuscript.

10. Lines 61 & Eq. 1: The spacing in "reference" is odd. Maybe this is just a quirk of latex. Did you use \mathit for the subscript.

We believe this was a quirk of LaTeX, but we have adjusted the spacing so that it looks more conventional.

11. Lines 75 and following: Please be a little more explicit about the units in these eqs. In particular, it requires some work for the reader to determine whether $\Delta(\delta APO)$ is a difference (in permeg) or a flux (in permeg/yr) of some other units.

We have clarified in the text preceding Eq. 4-5 that $\Delta(\delta APO)$ represents deviations of APO from the baseline expressed in per meg. To make this clear, we have adjusted how Eq. 4-5 are presented so that they i) match the derivation of Manning and Keeling (2006) and ii) use the same notation (i.e. changing $S_{O2}$ to $X_{O2}$). Where an additional term to correct for dilution effects of atmospheric $O_2$ is included, we remove this term as this was not used in the modelling of APO.

12. The citation of Pickers is incomplete.

This has now been corrected in the manuscript.

13. "Variations in …" is a run-on sentence. A new one should begin at "however".

This has now been corrected in the manuscript.

14. Line 144: Should read "for the influence of rapid variations in $CO_2$ flux on mole fractions, footprints are …"

This has now been corrected in the manuscript.

15. Figure 3 caption: I assume panel D shows the footprint for WAO. This is not stated in the caption.

We have specified in the caption that the footprint in Fig. 3d is for WAO.

16. Figure 4: In the figure caption, the distinction between groups of panels (a,b,c vs., d,e,f) is opaque. In my mind, there is no meaningful difference between "regional contribution" and "overall regional [contribution]". I believe you have just combined the four respective sets of tracers for $CO_2$ and $O_2$ in a,b,c (with appropriate weighting) to APO in d,e,f. The word "overall" does not convey this relationship. Please make this easier to discern from the caption.

We agree that as worded this is confusing. To make this clearer for the reader we have changed "*regional*" contribution to "*gas-specific sectoral*" contribution in the first instance. We have changed "*overall regional*" to "*APO*" in the second instance.

17. Line 320: "The APO model…". Exactly which model? By eye it's not obvious which lines have been included in the average, yielding the quoted $R^2$ value of 0.24 for December.

Here, "*APO model*" refers to all APO simulations that are presented rather than one specific simulation or subset of simulations that impose different data filtering. This has now been changed in the manuscript to now read "*all APO simulations*".

Please note, the $R^2$ and RMSE values presented in the text of the manuscript appear to be typos when compared to the Python code output for the mean $R^2$ and RMSE values of the APO simulations. The mean $R^2$ and RMSE for August 2015 (all simulations) should be: 0.1 and 8.4, respectively. The mean $R^2$ and RMSE for December 2015 (all simulations) should be: 0.3 and 7.1, respectively. We have corrected these in the main text.

For reference:

|  | August 2015 | | December 2015 | |
|---|---|---|---|---|
|  | $R^2$ | RMSE | $R^2$ | RMSE |
| All data | 0.09 | 8.98 | 0.27 | 7.01 |
| Daytime filtered | 0.07 | 8.81 | 0.36 | 6.22 |
| Ocean filtered | 0.14 | 7.49 | 0.36 | 8.12 |
| **Average** | **0.10** | **8.43** | **0.34** | **7.12** |

18. Figure 6 caption: Is there a superfluous "and" in the 3rd line?

This has now been corrected in the manuscript.

19. Line 340: The correspondence between $\alpha_f$ and $\alpha_b$ in panels of Fig. 7 is not as stated and it's not simple to state correctly, so just "the top and bottom panels of" and "respectively".

We agree that this was written in a confusing manner and have amended as suggested.

"… sensitivity of APO to $\alpha_b$ and $\alpha_f$ is shown in the top and bottom panels of Figure 7, respectively".

Now reads:

"… sensitivity of APO to $\alpha_b$ and $\alpha_f$ is shown in  Figure 7, ".

20. Table 1: The extension of the vertical line (in the column headings) separating August 2015 from December 2015 is in the wrong place.

This has now been corrected in the manuscript.

21. Line 413: Should "large-timescale" really be "long-timescale"?

This has now been corrected in the manuscript to read "long-timescale".

22. Figures 6 & 10: In the first case you simply say "$R^2$" and in the second you say "The Pearson correlation coefficient $R^2$". Either switch the order (giving a more complete introduction and subsequent abbreviation) or just stick with the simpler $R^2$ in both cases.

Thank you, this is a good point. We have removed "The Pearson correlation coefficient" from the caption of Fig. 10 and only use $R^2$ in both Fig. 6 and Fig. 10.
* * *
**Additional changes made by the authors.**

In addition to small changes in the wording of the manuscript text, that are detailed in the tracked changes document, the following changes of note have been made to the manuscript.

1. A typo on the left-hand-side term of Eq. 3 in the manuscript has been corrected from "APO" to "$\delta$APO".

2. We have adjusted Eq. (4-5) to match the derivation presented in Manning and Keeling (2006) and to use the same notation. We introduce Eq. (6), formerly Eq. (5), that includes the correction term for atmospheric $O_2$. This has been done to make it clear where this term of $1/(1-X_{O2})$ comes from as it is not included in the derivation presented in Manning and Keeling (2006)

3. Errata on line 45: The thesis of Kuijpers modelled atmospheric $O_2$ for autumn of **2014** and compared simulations with observations from **two** sites.

4. Figure 13. In the text this is described as "The correlations ($R^2$) between the observation-derived ffCO$_2$ and the ffCO$_2$ model" but the figure caption are in per meg, not ffCO2 in ppm. We have corrected the figure label and caption.

---

## Author Comment (AC2)

**"Atmospheric oxygen as a tracer for fossil fuel carbon dioxide: a sensitivity study in the UK" – Response to Anonymous Referee #4**

Hannah Chawner[1], Karina Adcock[2], Eric Saboya[1,7], Tim Arnold[3,4], Yuri Artioli[5], Caroline Dylag[3], Grant L. Forster[2,6], Anita Ganesan[7], Heather Graven[8], Gennadi Lessin[5], Peter Levy[9], Ingrid T. Luijkx[10], Alistair Manning[11], Penelope A. Pickers[2], Chris Rennick[3], Christian Rödenbeck[12], Matthew Rigby[1].

[1]School of Chemistry, University of Bristol, Bristol, UK
[2]Centre for Ocean and Atmospheric Sciences, School of Environmental Sciences, University of East Anglia, Norwich, UK
[3]National Physical Laboratory, Teddington, UK
[4]School of Geosciences, University of Edinburgh, Edinburgh, UK
[5]Plymouth Marine Laboratory, Plymouth, UK
[6]National Centre for Atmospheric Sciences, University of East Anglia, UK
[7]School of Geographical Sciences, University of Bristol, Bristol, UK
[8]Department of Physics, Imperial College London, London, UK
[9]Centre for Ecology and Hydrology, Edinburgh, UK
[10]Meteorology and Air Quality, Wageningen University and Research, Wageningen, the Netherlands
[11]Hadley Centre, Met Office, Exeter, UK
[12]Max Planck Institute for Biogeochemistry, Germany

Below we describe the changes that have been made to the re-submitted manuscript "Atmospheric oxygen as a tracer for fossil fuel carbon dioxide: ac sensitivity study in the UK" as recommended in the comments from the two reviewers.

Additional small changes that primarily relate to typos spotted by the authors have also been made and are listed towards the end of this document. Please note that these small additional changes made by the authors do not substantially affect the scientific findings.

Referee comments are stated in black with author responses in blue. Proposed changes to the manuscript are clearly stated throughout (usually in bold typeface).

**RC2 referee comments**

**General comments:**

1. The structure of the results is challenging to follow. Sections 3.1 to 3.5 discuss sensitivity results related to APO simulation, while Section 3.6 presents estimated $CO_2$. It's unclear whether the main focus of this work is on the robustness of the methodology or the $CO_2$ emission results. If the goal is to infer $CO_2$ emissions, why not directly use $CO_2$ emission results for sensitivity testing?

While we agree that the analysis is (necessarily, we feel) somewhat complex, we do not feel that it would be desirable to express everything in terms of $ffCO_2$ in these sections (note however, that this is the fossil fuel $CO_2$ mole fraction, not emissions). This is because: a) $ffCO_2$ is derived from APO, so we feel that is important to understand how this quantity underlying the $ffCO_2$ calculation is influenced by each factor; b) $ffCO_2$ requires an estimate of the background APO, which is one of our sensitivity tests.

2. Given the complexity of the study's methodology, it would be helpful to provide an overall workflow figure at the beginning to help readers better understand the process. Similarly, creating a table listing all sensitivity test settings could improve the readability of the sensitivity tests.

It is a good idea to include a table outlining the different sensitivity tests that have been carried out. We have added the following at L219:

"The sensitivity tests (for APO and $ffCO_2$) are summarised in Table 1.

Table 1: Summary of sensitivity tests. The left-hand column indicates the parameter being investigated and whether the sensitivity to APO or ffCO$_2$ is being investigated. The middle column briefly describes the method employed to determine the sensitivity, and the relevant results section is shown to the right."

| Sensitivity test | Method | Section |
|---|---|---|
| APO: Biosphere exchange ratio ($\alpha_B$) | Monte Carlo ensemble | 3.2 |
| APO: Fossil fuel exchange ratio ($\alpha_F$) | Monte Carlo ensemble Comparison of GridFED and NAEI-derived ratios | 3.2 |
| APO: Ocean flux estimate | Comparison of NEMO, ECCO-Darwin, Jena Carboscope flux estimates | 3.3 |
| APO: Fossil fuel flux magnitude and distribution | Monte Carlo ensemble Comparison of NAEI and EDGAR distributions | 3.4 |
| APO: Background | Comparison of JC and REBS | 3.5 |
| ffCO$_2$: Background and ocean flux estimate | Comparison of JC and REBS baseline Comparison of NEMO, ECCO-Darwin, Jena Carboscope ocean fluxes | 3.6 |

3. In my opinion, the usage of the term 'the regional contribution' may not be suitable for this study. Generally, 'regional contribution' refers to the portion or influence of a specific region on a particular phenomenon or variable. In this study, 'the regional contribution' is used to indicate contributions from ocean and fossil fuel components, which could lead to misunderstanding.

We do not believe that the reviewer is correct here. In this study, "regional contribution" does indeed correspond to the influence of fluxes from a particular geographical domain. It's just that we examine the influence of different sources (fossil, ocean, etc.) from within this region.

4. The selection of August and December as the study period should be explained and justified, especially when the other months, like June (with the lowest $R^2$) and November (with the highest $R^2$) as shown in Figure 6, might be more prominent.

Whilst there is good data availability of observations from Weybourne in 2015 (except during February) we found the balance of data availability, statistical goodness-of-fit, and having two months that represent sufficiently distinct parts of the APO seasonal cycle led to using August and December for the study period.

We have included the following at the start of Section 3:

"*Here we show our APO model results for 2015. As example, one summer (August) and one winter month (December) are shown throughout. **These months were selected based on data availability, statistical goodness-of-fit and having two months that represent sufficiently distinct parts of the APO seasonal cycle. Simulations for all months of …***"

5. The presentation of data in figures is quite simplistic, and there is a lack of standardization in the formatting of words inside the figures. For example, 'co2' should be written as 'CO$_2$.' It's necessary to review all figures and consider diversifying the ways data is presented.

We have updated the labelling in figures so that they are more standardized. We will take into consideration the way the data is presented (and opt for more diverse ways) in future work.

**Specific comments:**

6. Caption in Figure 1. What is fullname of UKGHG?

We have amended the caption in Fig. 1 to include the full name of the UKGHG flux model. Please note we have also corrected the reference from White et al. (2019) to Levy (2020), which is more appropriate.

7. Figure 4. Combine Figure 4, there is no need to split it across two pages.

We understand the reviewer's point of view regarding Fig. 4. However, having tried combining this figure onto one page makes it unclear for the reader to see the temporal variations of the different lines. On reflection, we think it is clearer to keep the figure as presented.

8. Figure 6. When using gray lines as major grid lines, I recommend that the author refrain from using gray lines for plotting the "no ocean" results. Please review all your figures to correct them.

We understand the reviewer's point of view regarding the line colours. However, whilst similar line colours are used for the "no ocean" results we have used a different line style to differentiate from the gridlines used in the plots.

9. Table 1. The table caption should be positioned above the table.
Thank you for spotting this, we have now moved the caption for Table 1 as suggested.
* * *
**Additional changes made by the authors.**

In addition to small changes in the wording of the manuscript text, that are detailed in the tracked changes document, the following changes of note have been made to the manuscript.

1. A typo on the left-hand-side term of Eq. 3 in the manuscript has been corrected from "APO" to "$\delta$APO".

2. We have adjusted Eq. (4-5) to match the derivation presented in Manning and Keeling (2006) and to use the same notation. We introduce Eq. (6), formerly Eq. (5), that includes the correction term for atmospheric $O_2$. This has been done to make it clear where this term of $1/(1-X_{O2})$ comes from as it is not included in the derivation presented in Manning and Keeling (2006)

3. Errata on line 45: The thesis of Kuijpers modelled atmospheric $O_2$ for autumn of **2014** and compared simulations with observations from **two** sites.

4. Figure 13. In the text this is described as "The correlations ($R^2$) between the observation-derived ffCO$_2$ and the ffCO$_2$ model" but the figure caption are in per meg, not ffCO2 in ppm. We have corrected the figure label and caption.